# Anatomy of subcritical submarine flows with a lutocline and an intermediate destruction layer

Jorge S. Salinas [1✉], S. Balachandar [1], M. Shringarpure[2], J. Fedele[2], D. Hoyal[2], S. Zuñiga[3,4,5] &
M. I. Cantero[3,4,5]

Turbidity currents are sediment-laden flows that travel over a sloping bed under a stagnant ambient fluid, driven by the density difference between the current and the ambient. Turbidity currents transport large amounts of carbon, nutrients and fresh water through oceans and play an important role in global geochemical cycling and seafloor ecosystems. Supercritical currents are observed in steeper slopes. Subcritical currents are observed in milder slopes, where the near-bed and interface layers are prevented from interacting across the velocity maximum. Past works show the existence of such a barrier to vertical momentum transfer is essential for the body of the subcritical current to extend over hundreds of kilometers in length without much increase in height. Here we observe the body of subcritical currents to have a three layer structure, where the turbulent near-bed layer and the non-turbulent interface layer are separated by an intermediate layer of negative turbulence production. We explain the mechanism by which this layer prevents the near-bed turbulent structures from penetrating into the interface layer by transferring energy back from turbulence to the mean flow.

[1] Department of Mechanical and Aerospace Engineering, University of Florida, Gainesville, FL 32611, USA. [2] ExxonMobil Upstream Research Company, Houston, TX 77389, USA. [3] Instituto Balseiro, Universidad Nacional de Cuyo, San Carlos de Bariloche, Argentina. [4] Centro Atómico Bariloche, Comisión Nacional de Energía Atómica, San Carlos de Bariloche, Argentina. [5] Consejo Nacional de Investigaciones Científicas y Técnicas, San Carlos de Bariloche, Argentina. ✉email: josalinas@ufl.edu

Turbidity currents are sediment-laden, gravity-driven underflows that travel down slope, and they are bounded by a sloping bed at the bottom and a layer of clear ambient fluid above[1,2]. The excess density of the current compared to the ambient fluid, due to suspended sediments, propels the current forward. In turbidity currents, fluid turbulence is the primary mechanism of retaining the sediments in suspension, which distinguishes them from debris flows. Turbidity currents are subdivided in terms of sediment concentration, into low and high density turbidity currents. Furthermore, they can be differentiated in terms of cohesive and non-cohesive sediments that they carry[3]. Turbidity currents are responsible for the formation of deeply eroded submarine canyons and channels that feed into giant deep-sea fans that represent the largest sedimentary accumulations on Earth. They transport large amounts of carbon, nutrients, and fresh water through oceans and therefore play an important role in global geochemical cycling, climate, and seafloor ecosystems[4]. They are responsible for the widespread emplacement of sediment as turbidites, which can contain large amounts of organic matter, and these deposits now form many oil and gas reserves[3,5]. Unlike the much studied problem of sediment transport by rivers, our current understanding of sub-aqueous turbidity currents is lagging, due to limited availability of direct field measurements and observations (e.g., refs. [6–10]). Much of our understanding is derived from interpretations of sediment deposits resulting from these flows, laboratory experiments that are necessarily limited to much smaller scale, and computer simulations that involve unavoidable assumptions and approximations.

In the bed-normal direction, a turbidity current can be broadly characterized by a near-bed layer, where the velocity increases from the no-slip condition at the bed to a maximum at the top of the layer, and an interface layer, where the velocity decreases from the maximum value back to zero at the boundary between the current and the ambient fluid. Along the length of the current, each flow event of a turbidity current is characterized by an energetic rapidly-varying front or head, followed by a long body in which the current slowly varies along the flow direction, and a short tail region that marks the end of the event. Depending on how prolonged the flow is, the turbidity current can be a surge-type with a prominent head followed by a short body and a tail, or a current that runs for days with a frontal region followed by a long body[7]. The bed-normal structure and the turbulent nature of the flow within the near-bed and the interface layers are different within the head, body, and tail regions of the current, and between the surge-type and long-running currents. For example, recent field measurements at the front of the current have provided valuable quantitative information on the structure of the head region, which consists of a dense near-bed layer that exchanges sediment with the bed via erosion and deposition[6,8,10]. At the head of the current, the interface layer is also highly turbulent resulting in rapid entrainment of ambient fluid and mixing with the current. On the other hand, it has been observed that along the body of a long-running current the sediment concentration is smaller than 1% and the current velocities are slower than 1 m s$^{-1}$ [7].

In this work, the focus is on the long running body of the current, where we assume the sediment concentration to be sufficiently dilute over the entire thickness of the current, without the presence of a dense near-bed layer. This assumption allows the use of Boussinesq model of the governing equations and also renders sediment-sediment interactions effects, such as hindered settling, negligible. Furthermore, we consider currents with either washload sediment, where the settling effects are negligible and the flow resembles conservative gravity currents driven by temperature or salinity differences, or non-cohesive sediment whose

settling velocity is significantly smaller than flow velocity. Finally, we assume the body of the turbidity current to be in bypass mode, where sediment erosion and deposition occur along the bed, but their rates nearly balance each other so that the streamwise flux of suspended sediment is constant along the length of the current. These assumptions give the body of the turbidity current a specific slowly varying character, such as the body of the type 1 events observed in Simmons et al.[10]. It must be cautioned that the structure and the dynamics of the body of the current can differ under conditions of strong net erosion or deposition.

Here we seek to understand the structure of the body of a subcritical current, and its ability to evolve downstream without significant mixing with the ambient fluid. It must be emphasized that the subcritical turbidity current must remain turbulent in the near-bed region, for otherwise it would not be able to keep the settling sediments in suspension. Nevertheless, this near-bed turbulence is prevented from encroaching upward into the interface layer with the lutocline acting as a flowing fluid lid. Furthermore, density stratification in the interface layer is maintained stronger than local velocity gradient to suppress instabilities and maintain the layer non-turbulent. In this work, we present a three-layer structure for the body of a subcritical turbidity current, where an intermediate layer allows the interface layer to remain free of turbulence. Previous works have addressed the separation of the near-bed and interface layers—Buckee et al.[11] proposed a minimum in turbulence production and a barrier to momentum transfer near the streamwise velocity maximum. Luchi et al.[12] argued that such a barrier could explain the lack of mixing near the upper boundary. Recently, Dorrell et al.[13] reported self-sharpening of velocity and concentration profiles as mechanisms responsible for the formation of a stable barrier to mixing that can be linked to the long runout of gravity currents in the Black sea. Expounding on the multi-layered structure of a subcritical current and offering a mechanistic picture of its inner workings is the primary goal of this study.

## Results and discussion

**The subcritical regime**. In turbidity currents, the excess density of the current over that of the ambient fluid plays a unique dual role. On the one hand, the excess density is the only source of streamwise momentum and thus is solely responsible for flow turbulence. On the other hand, the stable vertical density gradient from the bottom of the current to the ambient fluid above tends to damp turbulence. The balance between the two competing mechanisms results in two distinct flow states in the body of the current, which are described as supercritical and subcritical. The balance between inertial and buoyancy effects is determined by the densimetric Froude number $\mathrm{Fr} = U/(Ch/\tan\theta)^{1/2}$, where the depth-averaged mean streamwise velocity $U = \int_0^\infty \overline{u}^2 \, dz / \int_0^\infty \overline{u} \, dz$ and the depth-integrated net suspended sediment $Ch = \int_0^\infty \overline{c} \, dz$[14,15], with $\theta$ being the slope of the bottom bed, $\overline{u}(z)$ and $\overline{c}(z)$ being the average streamwise velocity and sediment concentration profiles, respectively. Supercritical currents ($\mathrm{Fr} > 1$) are observed in steeper slopes where the turbulence production mechanism dominates over the damping effect. As a result, both the near-bed and the interface layers are turbulent in a supercritical current[16]. Subcritical currents ($\mathrm{Fr} < 1$) are observed in milder slopes[17] where the damping effect of stable density stratification in the interface layer dominates over shear production of turbulence. As a result, along the body of a subcritical current (away from the head), only the near-bed layer is turbulent, while the interface layer remains free of turbulence[18]. The sediment concentration remains well mixed within the near-bed layer and rapidly decreases to zero across the stably-stratified interface layer, giving the appearance of the current and the

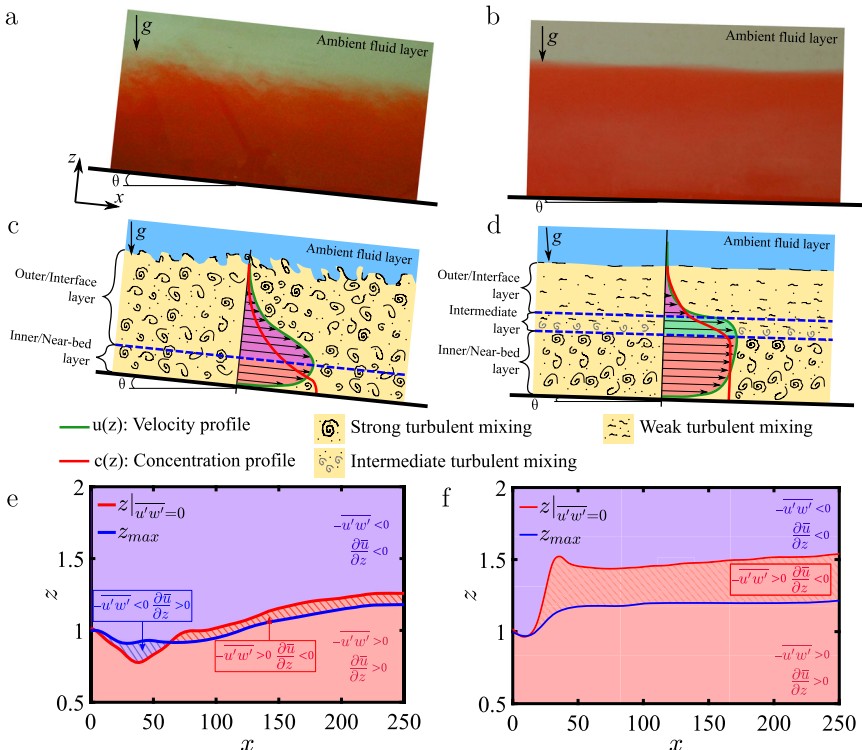

**Fig. 1 Structure of supercritical and subcritical currents.** Saline flow laboratory experiment of: **a** dilute supercritical current, height $h = 0.05$–$0.08$ m, Fr = 1.6–1.98, Re = 18,000–20,000; **b** dilute subcritical current, height $h = 0.1$–$0.13$ m, Fr = 0.1–0.12, Re = 6000–9000. Schematic representations of the body of: **c** a dilute supercritical current; **d** a dilute subcritical current. Non-dimensional bed-normal location of streamwise velocity maximum $z_{max}$ (solid blue line) and zero Reynolds stress $z|_{\overline{u'w'}=0}$ (solid red line) as a function of downstream location $x$ for the numerical simulations of: **e** a dilute supercritical gravity current; **f** a dilute subcritical gravity current.

ambient being immiscible (i.e., as if the current is capped by a lid). This region of very strong density gradient is called lutocline, which has been observed in many natural flows and laboratory turbidity and gravity currents[18,19].

Figure 1a, b shows two laboratory experiments of saline, gravity currents that reveal the difference between a supercritical current with a mixing turbulent interface (Fig. 1a) versus a subcritical current topped by a lutocline (Fig. 1b). The pictures show the body of a laboratory turbidity current in bypass mode, after the energetic head of the current has passed[20]. In many respects, the supercritical current resembles a turbulent wall-jet (TWJ)[21,22] (see Fig. 1c), where the two layers have also been referred to as the inner and outer layers[23]. However, there are differences between the supercritical current and TWJ due to presence of stable stratification in the former. The fascinating aspect of the subcritical current is the abrupt transition from a region of turbulence in the lower parts of the current to a thin non-turbulent stable lutocline (ref. [13]; see also Fig. 1d).

The existence of a stable lutocline on top of a subcritical current is of particular importance in many geophysical flows. Classical jets and plumes increase in thickness as they flow downstream by entraining and mixing with the ambient fluid, and as a result they eventually dilute themselves to extinction. In contrast, it has been observed that turbidity currents can have long runouts of over thousands of kilometers within submarine channels[7,10,24–26]. Surge-type turbidity currents in erosional mode can travel long distances and not dilute themselves to extinction due to the continuous supply of sediment from the bed. On the other hand, long running turbidity current events that span over several days imply currents with a very long body that extend along the submarine channel[7,10]. It can be argued that if the currents were to entrain and mix with the ambient

fluid and grow in height along the body of the current, they could not have extended over hundreds of kilometers in length, since this would result in impossibly thick currents. The ability of the body of a subcritical turbidity current to remain coherent over extended lengths has been considered in the past[12,24]. Reynolds averaged simulations of Luchi et al.[12] demonstrated the possibility of a very long body, provided turbulent mixing is suppressed in the eddy viscosity model at the velocity maximum. The present work will explain the mechanistic details of how turbulence transport is suppressed near the velocity maximum with fully-resolved simulations, without the use of any turbulence closure model. Furthermore, the present results will highlight the process by which turbulence production becomes negative in the intermediate destruction layer and thereby turbulent kinetic energy gets transformed back into mean flow kinetic energy.

A surge-type current, even though its length along the channel at any given time may not be long, can travel over a long runout since the head of the current is sufficiently energetic and net-erosional. Here, in contrast, we are interested in turbidity currents with a very long body along the submarine channel. The three-layer structure of the subcritical current and the resulting very slow bed-normal growth of the current are important features that offer a plausible mechanism for the existence of a long body in such long running currents.

**Key ingredients.** The present results will establish the three key ingredients that are necessary for the body of a subcritical current in bypass mode to evolve along the streamwise direction without growing in thickness: (M1) the near-bed layer must behave like a turbulent open channel flow (TCF) with a free-slip lid at the top,

(M2) the interface layer must remain turbulent free and thus exhibit weak diffusional growth, and (M3) there must be a stabilizing intermediate layer between the near-bed and interface layers that strongly suppresses any upward transport of near-bed turbulence into the upper interface layer. There is substantial understanding and support for each of the three ingredients in the literature, which will be discussed in the following paragraphs. However, much of these understandings are independent of each other and in different contexts. In this work, we will integrate these understandings into a coherent description of the lutocline.

In a turbulent wall jet (TWJ), the near-bed layer grows as a turbulent boundary layer (TBL) and the interface layer grows as free-shear layer (FSL). In the infinite Reynolds number limit, the near-bed (inner) and interface (outer) layers can be taken to be independent[27]. At finite Reynolds numbers, the near-bed and interface layers cannot be taken to be independent and the higher intensity of turbulence in the interface layer results in the intrusion of the interface layer into the near-bed layer[23,28]. Nevertheless, at all Reynolds numbers, both the near-bed and the interface layers contribute to the continuous thickening of the TWJ. We note that the body of a turbidity current in bypass mode is qualitatively similar to a turbulent wall jet (TWJ), except for the added effects of density stratification. Thus, the focus here is to identify the key features that distinguish a subcritical current from a TWJ.

M1: It has now been established that the turbulent length scale in a TWJ is substantially larger than in a turbulent open channel flow[29], and this increase is primarily due to the interaction between the near-bed and interface layers. Recent high quality experiments and simulations of TWJs have revealed the structure of the two layers and their interaction[30,31]. Two turbulent production peaks were observed, one in each layer, with the peak in the interface layer being an order of magnitude larger than in the near-bed layer. As a result, in TWJs, turbulence from the interface layer is transported into the near-bed layer by triple velocity correlations[30]. In turbidity currents, the interaction between the two layers is modified by the presence of stable density stratification. We will show that, in the case of a supercritical current, interaction between the near-bed and interface layer is observed, however, the nature of interaction is reversed with turbulence from the near-bed layer being transported to the interface layer. As a result, though the near-bed layer of a supercritical current grows downstream, this growth is far lower than in a turbulent wall jet. In a subcritical current, due to further enhancement of the stabilizing effect of density stratification, it can be conjectured that the nature of near-bed turbulence is similar to that of a turbulent open channel and the growth of the near-bed layer is nearly halted.

M2: The condition for stability of a stratified shear layer is given in terms of gradient Richardson number as $Ri_g > 0.25$[32,33], where gradient Richardson number $Ri_g = -(\partial \bar{c}/\partial z)/(\tan \theta (\partial \bar{u}/\partial z)^2)$. In applying this condition to the interface layer of a turbidity current, it should be recognized that neither the amount of shear (denominator) nor the magnitude of stable stratification (numerator) is externally imposed. They are internally determined by the partitioning of suspended sediment between the near-bed and the interface layers, and by the maximum velocity attained by the current within the near-bed layer. The stability condition is not satisfied in the interface layer of a supercritical current and the upper layer is observed to remain turbulent. The stability condition is satisfied everywhere in the upper layer of a subcritical current and we observe the layer to be non-turbulent. Although several studies have focused on the stable lutocline layer[19,34–37], many aspects of its internal structure and its detailed interaction with the turbulent near-bed layer remains to be explored.

M3: The behavior of near-bed vortical structures as they ascend through the log layer and approach the location of the velocity maximum and the upper interface layer is of particular interest. We hypothesize the existence of a substantially thick layer of negative turbulence production between the turbulent near-bed and interface layers. In this intermediate layer of destruction, turbulent fluctuations are actively converted back to mean flow variation. A narrow region of negative production has been recognized in TWJs and gravity currents in the region of velocity maximum[30,31]. The inability of near-bed turbulence to promote instability in the interface layer can also be explored on the basis of interaction between the turbulent vortical structures and the stably-stratified layer. In a TBL, the outer turbulence peak that increases in intensity with increasing Reynolds number is linked to the hairpin packets freely reaching into the log region of the near-bed layer[11,31,38]. On the other hand, it has been demonstrated that a layer of strong enough stratification behaves like a slip wall blocking the passage of vortices. Furthermore, the presence of negative turbulent production has been associated with coherent vortices that are inclined in the direction of shear[39]. The role of negative turbulence production within the intermediate layer and its structural origin in the form of coherent inclined vortices and their relation to near-bed vortical structures will be explored in detail.

**Simulation details**. Results from highly-resolved direct numerical simulations of turbidity currents flowing down a bed of slope $\theta$ will be used to gain deeper insight. Our numerical simulations model a streamwise segment of the long body of the turbidity current, away from the energetic front and the weak tail. Therefore, the head of the current that forms at the beginning is allowed to travel downslope and exit the computational domain. The long body of the current that remains within the computational domain after this initial transient phase is investigated in detail, with particular attention to the nature of the interface layer and the entrainment of ambient fluid from above. The simulations consider dilute concentration of sediment within the body of the current, which allows Boussinesq approximation in the governing mass and momentum balance equations. By restricting to the dilute body of the current, the present simulations do not model the dense near-bed layer observed near the front of the current in recent field measurements[6–8,10]. Suspended sediment is assumed to be non-cohesive and sufficiently small in size that its settling velocity can be either ignored in comparison to the flow velocity (i.e., sediment treated as washload) or can be taken to be the sum of local fluid velocity plus still fluid settling velocity of the sediment (i.e., equilibrium Eulerian approximation[40–42]). We also ignore the effect of hindered settling on account of low sediment concentration.

Under these conditions, the conservation equations of fluid mass, momentum and sediment concentration are as given in Salinas et al.[43,44]. The results to be discussed are non-dimensionalized with the half-height $H$ of the current at the inlet as the length scale, average concentration $c_v$ at the inlet as the concentration scale, and $u_* = \sqrt{g' \sin \theta H}$ as the velocity scale[43]. Here $g' = R c_v g$ is the reduced gravity with $g$ being the acceleration due to gravity, and $R = \rho_s/\rho_f - 1$, where the density of sediment and clear fluid are $\rho_s$ and $\rho_f$, respectively. The dimensionless parameters are the shear Reynolds number $Re_\tau = u_* H/\nu$ and the Schmidt number $Sc = \nu/\kappa$, which are chosen to be $Re_\tau = 180$ and $Sc = 1$. Here $\nu$ is the kinematic viscosity and $\kappa$ is the sediment diffusivity. The bulk Reynolds number $Re = \int_0^\infty \bar{u}\, dz/\nu$ of the resulting flow within the body of the current ranged from 6000 at inlet to 12,000 at outlet of the computational

domain. The dimensionless sediment settling velocity is defined as $V = R g d^{*2}/(18 \nu u_*)$, where $d^*$ is the dimensional particle diameter.

Among the many simulations performed, results from three particular ones will be highlighted: (i) subcritical gravity current with a bottom slope $\theta = 0.29°$, inlet densimetric Froude number Fr $= 0.83$, and washload sediment of zero settling velocity $V = 0$, (ii) supercritical gravity current with $\theta = 2.86°$, Fr $= 2.65$, and $V = 0$, and (iii) subcritical turbidity current with $\theta = 0.29°$, Fr $= 0.83$, and non-dimensional settling velocity $V = 10^{-3}$. As an example, we now place the above subcritical turbidity current in physical terms, first in the context of a laboratory experiment and then in the context of a possible field condition. For laboratory experiments similar to those of Sequeiros et al.[20] with sediments of specific gravity 1.53, consider a subcritical turbidity current whose body is of height 22.8 cm, driven by sediment of size ~ 7 μm at a volumetric concentration of 4% down a slope of $\theta = 0.29°$. The mean velocity of the resulting current is 0.19 m s$^{-1}$, which yields a bulk Reynolds number of 34,369, and a non-dimensional sediment settling velocity of 10$^{-3}$, which are fully consistent with the simulation parameters. In the context of field scale flow, where the specific gravity of sediments is 2.65, consider a dilute subcritical turbidity current of height 48.4 m along the body of the current driven by 8 μm sediments at a volumetric concentration of 0.1%. The resulting mean flow velocity is about 0.8 m s$^{-1}$ yielding again $V = 10^{-3}$. The bulk Reynolds number at field conditions is, however, much higher at $2.97 \times 10^7$. These conditions are well within the range of values reported in the field measurements of Xu et al.[45], Azpiroz-Zabala et al.[7] and Simmons et al.[10]. The non-dimensional settling velocity $V$ scales as square of sediment size and therefore sediments of even smaller size can be considered as washload with $V \approx 0$. Furthermore, consider a dilute supercritical turbidity current flowing down a sloping bed of $\theta = 2.86°$, with height 17.3 m along the body, driven by sediment of size 11 μm, specific gravity 2.65 and at a volumetric concentration of 0.1%. This results in a mean velocity of 1.17 m s$^{-1}$ and bulk Reynolds number of $1.56 \times 10^7$.

In all the simulations, the body of the current is assumed to be in the bypass mode, where the net exchange of sediments between the bed and the current is set to zero. This assumption implies that the rate of resuspension of sediments from the bed is equal to the rate at which sediments settle onto the bed[43,44]. The advantage of this assumption is that it greatly simplifies the boundary condition to be applied at the bottom of the computational domain and renders the streamwise sediment flux to be a constant along the streamwise segment of the body of the current being simulated. The simulations were performed in a computational domain of streamwise length 150 times the height of the current at the inlet, which under field conditions quoted above correspond to a streamwise segment of length 7.3 km. A turbidity current with a roof[46] enters the domain at the left as inflow and convective boundary conditions are applied at the right boundary of the computational domain. The spanwise extent of the computational domain is taken to be 4 times the height of the current at the inlet, which along with periodic boundary condition corresponds to a channel of width larger than 194 m.

**Two- and three-layer structure.** First we establish the existence of a three-layer structure in the case of a subcritical current in contrast to the dominant two-layer structure of a supercritical current. Figure 2 shows span-averaged concentration field $\bar{c}$ of the numerical simulations of (a) supercritical and (b) subcritical currents. Note that the figures are stretched in the bed-normal direction for better visualization. Stratification is strong in the body of the subcritical current resulting in a stable interface with a lutocline of rapid density variation. On the other hand, strong interfacial mixing is present in the supercritical current, which can be observed in the contours of constant concentration $\bar{c} = 0.01$ shown in yellow as the upper edge of the current. As shown in the schematic of Fig. 1c, the body of the supercritical current presents a nose-down, TWJ-like velocity profile, with a monotonically decreasing concentration profile that reaches an almost constant concentration near the bottom boundary[43,44,47]. On the other hand, the body of the subcritical current is characterized by a nose-up velocity profile and a nearly constant concentration of sediment below the velocity maximum, capped by a lutocline above it (see Fig. 1d). The structure of the subcritical current consists of three distinct layers: a near-bed layer of strong turbulence bounded between the bottom bed and the velocity maximum (bottom dashed blue line in Fig. 1d); an intermediate destruction layer, where turbulent kinetic energy (TKE) production is negative and turbulence is actively converted back to mean flow; a stably-stratified interface layer delimited by its border with the intermediate layer marked as the dashed blue line in Fig. 1d. On the other hand, the body of the supercritical current (Fig. 1c) is effectively comprised of only the near-bed and the interface layers.

The intermediate layer is identified as the region of negative total turbulent kinetic energy (TKE) production. Here TKE is

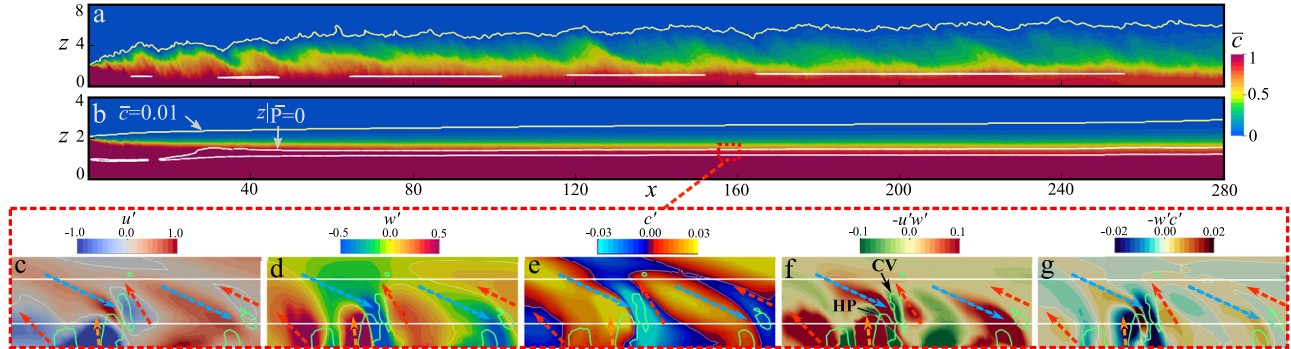

**Fig. 2 Spanwise averaged concentration $\bar{c}$ and perturbations from the mean at $y = 1.95$.** Spanwise averaged concentration field $\bar{c}$ for numerical simulations of: **a** dilute supercritical and **b** dilute subcritical gravity currents along the body of the current (away from the head). Yellow contours for $\bar{c} = 0.01$ indicates the interface between the current and the ambient layer. White contours correspond to zero total turbulent kinetic energy (TKE) production. Closeups for the subcritical current show: **c** $u'$; **d** $w'$; **e** $c'$; **f** Reynolds stress $-u'w'$; **g** Bed-normal Reynolds flux $-w'c'$ at the plane $y = 1.95$ in the intermediate layer. Blue dashed arrows: $u' > 0$, $w' < 0$, $c' > 0$. Red dashed arrows: $u' < 0$, $w' > 0$, $c' < 0$. Orange dashed arrows: $u' < 0$, $w' > 0$, $c' > 0$. HP hairpin vortex, CV counter-clockwise rotating vortex.

defined as $\overline{k} = (1/2)\overline{u'_i u'_i}$, where the overbar ($\overline{\cdot}$) denotes span-time average, and prime $(\cdot)'$ denotes perturbation. TKE production is defined as $\overline{P} = -\overline{u'_i u'_j}\frac{\partial \overline{u}_i}{\partial x_j}$. The dominant contribution to TKE production arises from shear production, which is defined as the product of Reynolds shear stress and the mean velocity gradient as $\overline{P}_s = -\overline{u'w'}\partial\overline{u}/\partial z$. In Fig. 1e, f the location of maximum streamwise velocity, $z_{max}$, is shown as the solid blue line for our numerical simulations, below (and above) which the mean velocity gradient is positive (and negative). The location of zero Reynolds stress is plotted in these figures as the solid red line. The disparity in the locations of the velocity maximum and the zero Reynolds stress is indicative of the asymmetry of the mean velocity profile around the velocity maximum. Because of this disparity, in both the supercritical and subcritical currents, there exists an intermediate region above or below the velocity maximum where Reynolds shear stress ($-\overline{u'w'}$) and mean velocity gradient ($\partial\overline{u}/\partial z$) are of opposite sign resulting in negative shear production. In the supercritical current, near the inlet ($x \lesssim 60$) streamwise momentum dominates stratification and negative shear production is below the velocity maximum (striped blue region) and this scenario is similar to that observed in TWJs[30,31]. As stratification starts to dominate ($x \gtrsim 60$) the influence of the near-bed layer on the interface layer dominates, and the region of negative shear production switches to above the velocity maximum (striped red region). On the other hand, in the subcritical current the stratification effect dominates right from the inlet and the region of negative shear production is substantial.

It must be stressed that an intermediate layer near the velocity maximum where shear production is negative is a general property of all shear flows exhibiting an asymmetric velocity maximum. A barrier to momentum transport exists at the velocity maximum even in supercritical currents. Although shear production is negative at all streamwise locations in both the subcritical and supercritical currents, this is not the case for total TKE production, due to other contributions to turbulent kinetic energy. The region where total TKE production is negative is enclosed by the white contours in Fig. 2a, b. In the subcritical current, a substantial layer of fluid where total TKE production is negative clearly separates the near-bed turbulent region from the interface layer. The properties of this layer is further examined in Frames c to g. In contrast, in the supercritical case shown in Fig 2a, the white contours cover a negligible area within them and are discontinuous, indicating that total TKE production is virtually positive everywhere.

**Anatomical structure of a subcritical current.** To better understand the three-layer structure of a subcritical current we present in Fig. 3a the velocity and concentration profiles in the self-similar body of the current (far from inlet, $x > 120$) for the subcritical gravity (solid profiles) and turbidity currents (dash-dotted profiles), together with experimental data from Sequeiros et al.[20] and field data from Dorrell et al.[13]. The profiles are scaled by $U$ (bulk streamwise velocity), $C$ (bulk concentration), and $z$ by $h$ (current height)[14]. The horizontal dash-dot lines show locations of zero total TKE production with total TKE production being negative in the region $0.62 \lesssim z/h \lesssim 0.74$. For the small sediment size considered, the results of the subcritical turbidity current with non-zero sediment settling velocity are nearly identical to those of the simulation where the settling velocity is taken to be zero. Below $z/h \lesssim 0.45$ (region where $Ri_g < 0.25$), concentration is mainly constant and above concentration rapidly decreases, forming a lutocline. Good agreement can be seen between the numerical simulations and laboratory experiments[20] and fair agreement is observed with the field data of subcritical gravity

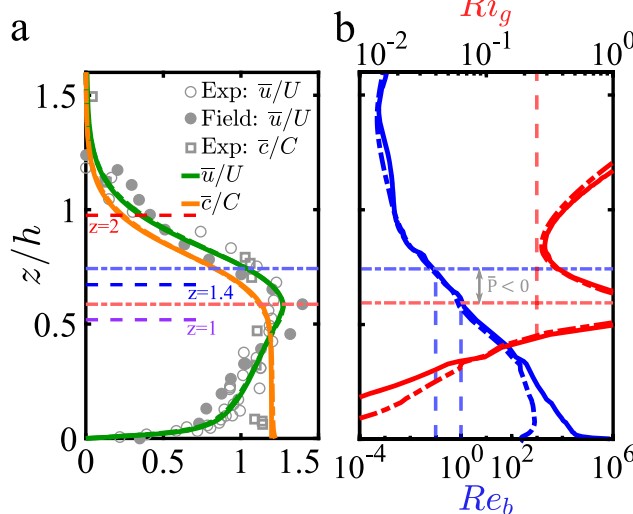

**Fig. 3 Scaled profiles as a function of scaled bed-normal location. a** Scaled streamwise velocity $\overline{u}/U$ and concentration $\overline{c}/C$ as a function of scaled bed-normal location $z/h$ for the subcritical gravity (solid profiles) and turbidity (dash-dotted profiles) currents. The profiles for both gravity and turbidity currents are almost identical. Also, we show experimental data from Sequeiros et al.[20], and field data from Dorrell et al.[13] of subcritical currents; **b** Buoyancy Reynolds $Re_b$ and gradient Richardson number $Ri_g$ as a function of scaled bed-normal location $z/h$ for the subcritical gravity (solid profiles) and turbidity (dash-dotted profiles) currents.

currents[13]. Also shown in Fig. 3b is buoyancy Reynolds number (blue profiles) $Re_b = Re_\tau \tan\theta \, \overline{\varepsilon}/(\partial\overline{c}/\partial z)$, where $\overline{\varepsilon} = \frac{1}{Re_\tau}\overline{\frac{\partial u'_i}{\partial x_j}\frac{\partial u'_i}{\partial x_j}}$ is the TKE dissipation. In the gravity current, $Re_b$ decreases six orders of magnitude from the bottom bed to the velocity maximum, where $Re_b$ becomes unity, while it decreases two orders of magnitude in the turbidity current. For both cases, buoyancy Reynolds number decreases one order of magnitude in the intermediate layer. In the interface layer, buoyancy Reynolds number decreases to values between $10^{-4} < Re_b < 10^{-3}$. Thus, buoyancy Reynolds number is consistent with the turbulent nature of the near-bed layer and the damped state of the interface layer. Also shown is the gradient Richardson number $Ri_g$ (red profiles), which with a value larger than 0.25 (vertical dash-dotted red line) corroborates hindered mixing at the interface region.

Figure 4a shows a composite plot of the subcritical gravity current in the fully developed region ($125 < x < 200$). Turbulent structures captured by isosurfaces of swirling strength[48] ($\lambda_{ci} = 10$) are colored by bed-normal location $z$, together with the region of negative total TKE production marked between the light blue and pink planar surfaces. The near-bed layer of the current is populated by forward-leaning hairpin vortices with their heads reaching just below the intermediate layer. In the intermediate destruction layer, a dilute distribution of weak counter-clockwise rotating vortices can be seen above the pink surface, identified by iso-surfaces of $\lambda_{ci} = 3.5$ (in green). They are induced by the clockwise rotating hairpin heads in the near-bed layer reaching into the intermediate layer from below. Figure 4b presents a closeup view of one of these structures colored by the bed-normal location $z$.

The rotation of these structures is better visualized in a contour plot of spanwise vorticity $\Omega_y$ on a plane going through the middle of these structures (see Fig. 4b). The clockwise hairpin vortex (denoted HP) that grew in the near-bed layer is just poking into the intermediate layer of negative production, but is unable to penetrate due to density gradient. It, however, induces the weak

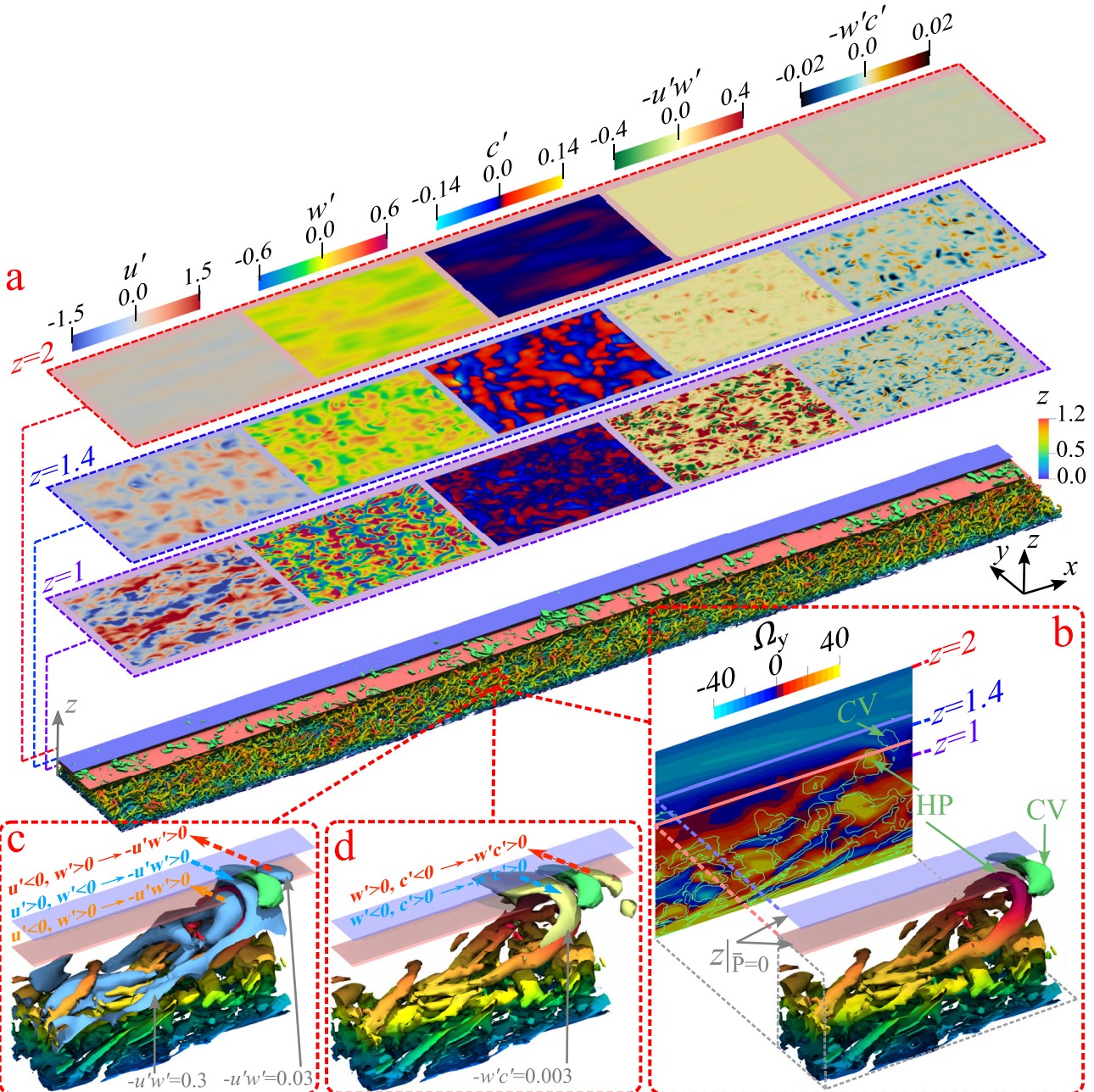

**Fig. 4 Composite plot of the subcritical gravity current along the self-similar body (125 < x < 200). a** Turbulent structures in the near-bed layer are captured by an iso-surface of swirling strength ($\lambda_{ci} = 10$) and colored by bed-normal location $z$, together with bottom and top isosurfaces where total TKE production is zero (light blue and pink surfaces). Turbulent structures in the intermediate layer are captured by an iso-surface of $\lambda_{ci} = 3.5$ (in green). Also, contours of perturbations from the mean ($u'$, $w'$, $c'$) and cross-correlations ($-u'w'$, $-w'c'$) at $z = 1$ (near-bed layer), $z = 1.4$ (intermediate layer) and $z = 2$ (interface layer) are shown. **b** Close-up of interaction between hairpin vortex "HP" and counter-clockwise rotating vortex "CV", together with planes of zero total TKE production $z|_{\bar{P}=0}$ and contours of spanwise vorticity $\Omega_y$ at $y = 1.95$ (plane through both the middle of structures HP and CV); **c** Structures HP and CV with iso-surfaces of constant Reynolds stress $-u'w' = 0.3$ and $0.03$ (light blue); **d** structures HP and CV with iso-surface of constant bed-normal Reynolds flux $-w'c' = 0.003$ (light yellow).

counter-clockwise rotating vortex (denoted CV) above and upstream (to the left in the figure) of it within the intermediate layer. Only the strongest of the clockwise rotating near-bed vortical structures that are able to poke into the intermediate layer are able to induce the counter-clockwise vortices within the intermediate layer. This explains the dilute distribution of weaker counter-rotating vortices within the intermediate layer, as can be seen in Fig. 4a.

**Structural origin of negative TKE production.** In most regions of turbulent flow, the mean velocity gradient and the Reynolds shear stress are of the same sign resulting in positive shear TKE production. An inclined vortex pair—top section of the hairpin (HP) and the weak counter-clockwise vortex (CV)—presents a classic coherent structure that contributes to negative turbulence production or counter-gradient transport of momentum[39,49]. The spatio-termporal persistence of these random distribution of

inclined vortex pairs is responsible for the sustained negative total TKE production within the intermediate layer.

Further insight can be gained by looking at the contours of perturbation velocity and concentration ($u'$, $w'$, $c'$), Reynolds stress $-u'w'$, and Reynolds flux $-w'c'$ on a vertical plane passing through the middle of the turbulent structures presented in Fig. 4b. These contours are shown in Fig. 2c–g, where the bed-normal axis has been stretched for better visualization and only the region around the intermediate layer is shown. By carefully choosing the contour levels to highlight the small variations seen in this region, we clearly identify two types of inclined structures: regions of positive streamwise velocity $u'$ correlated with negative bed-normal $w'$ perturbation (blue dashed arrows), and regions of negative streamwise velocity correlated with positive bed-normal perturbation (red and orange dashed arrows). In Fig. 2f, both these regions can be identified as inclined regions of positive Reynolds stress ($-u'w' > 0$). Also plotted in these frames are the hairpin and the induced counter-clockwise vortices (green contour of $\lambda_{ci} = 3.5$).

The three-dimensional nature of this vortex interaction and positive Reynolds stress is illustrated in Fig. 4c, where the HP and CV vortex structures identified in Fig. 4b are plotted along with iso-surfaces of positive Reynolds stress $-u'w' = 0.3$ and 0.03 (light blue). We find a region of positive $-u'w' = 0.3$ resulting from $u' > 0$ and $w' < 0$ (see blue dashed arrow) in between the head of the hairpin HP and the vortex CV. Moreover, positive Reynolds stress is observed in (i) the region downstream of the vortex CV (red dashed arrow, $-u'w' = 0.03$) and (ii) in between the legs of the hairpin (orange dashed arrow, $-u'w' = 0.3$), below the head of the hairpin in the ejection region, where positive Reynolds stress results from $u' < 0$ and $w' > 0$. Negative TKE production implies counter-gradient transport of momentum, which leads to transfer of energy from turbulent fluctuations to the mean streamwise shear flow. Experiments of subcritical gravity currents[11] have shown a region of negative shear production above the velocity maximum, and suggested the possible role of vortices. With the present analysis, it is clear that the inclined patches of positive Reynolds stress, which correspond to negative TKE production driven by the negative velocity gradient in this region, are due to the vortex pair HP and CV[39,49].

**Counter-gradient concentration transport.** An important point to note is that counter-gradient transport of momentum does not guarantee counter-gradient transport of concentration, which is an important mechanism by which the sharpness of the lutocline is maintained over long distances. In the near-bed and interface layers, high concentration parcels of fluid are transported upward ($c' > 0$ and $w' > 0$) and lower concentration parcels are transported downward ($c' < 0$ and $w' < 0$) tending to reduce concentration gradient. On the other hand, in the intermediate layer, positive regions of Reynolds flux $-w'c'$ are observed in Fig. 2g. These regions of positive Reynolds flux correlate well with the regions of positive Reynolds stress and are the result of the HV–CV vortex pair (blue and red dashed arrows). In these regions of positive Reynolds flux, parcels of high concentration are transported downward, while parcels of low concentration are transported upward and thereby enhancing concentration gradient in the intermediate layer. This is an essential feature of the self-sharpening of the concentration profile into a lutocline. A 3D view of the positive Reynolds flux events can be seen in Fig. 4d, where vortex structures HP and CV are presented along with iso-surfaces of $-w'c' = 0.003$ (light yellow). Counter-gradient transport of concentration occurs in between the head of the hairpin HV and the counter-clockwise vortex CV (blue dashed arrow) and in the region downstream of the vortex CV (red dashed arrow). However, negative values of Reynolds flux are found in the ejection region of the hairpin (below the head in between the legs, see orange dashed arrow in Fig. 4c).

To further explore how mixing is hindered at the interface layer we present in Fig. 4a blown up views of different quantities at three different bed-normal locations: below the velocity maximum ($z = 1$), in the region of negative total TKE production ($z = 1.4$) and in the interface layer ($z = 2$). These locations are also shown in Figs. 3a and 4b. We show perturbation streamwise and bed-normal velocities ($u'$ and $w'$), concentration ($c'$), Reynolds stress $-u'w'$, and Reynolds flux $-w'c'$. Note that the corresponding color maps are scaled by the values in the intermediate plane $z = 1.4$. Below the velocity maximum ($z = 1$), velocity and concentration perturbations are well correlated, which is reflected in the contours of Reynolds stress and fluxes as well. In the intermediate destruction layer, concentration fluctuations $c'$ are much larger than elsewhere and this is where the concentration gradient takes large values. As a result, Reynolds flux $-w'c'$ is the highest in the intermediate layer. Finally, in the interface layer ($z = 2$) where gradients of velocity is high and stratification hinders mixing, we still see non-negligible large-scale perturbations of velocity and concentration. However, these perturbations are un-correlated, as evidenced by the contours of Reynolds stress and flux. It is important to stress that, as indicated by the similarity of profiles in Fig. 3 and the ones to be presented in Fig. 5, the above results on the three-layer structure and the inclined vortex dynamics within the intermediate destruction layer remain virtually the same for both the subcritical currents of $V = 0$ and $V = 10^{-3}$.

**Implications.** Following the works of Parker et al. and others[15,30,50], the dimensionless mean streamwise momentum in the statistically stationary state simplifies to

$$\frac{1}{\mathrm{Re}_\tau u_\tau^3} \left( -\frac{\partial \overline{p}}{\partial x} + \frac{1}{\mathrm{Re}_\tau} \frac{\partial^2 \overline{u}}{\partial z^2} - \frac{\partial}{\partial z} \overline{u'w'} + \overline{c} - \frac{\partial \overline{u}^2}{\partial x} - \frac{\partial \overline{uw}}{\partial z} \right) = 0, \quad (1)$$

where $u_\tau$ is the local shear velocity computed from the bed-normal gradient of streamwise velocity at the bed. Figure 5a shows the terms of this balance as a function of bed-normal position $z^+ = \mathrm{Re}_\tau u_\tau z$ for the subcritical gravity current. Vertical dashed gray lines correspond to locations of zero total TKE production, which separate the three layers of the current. The suspended sediment through the term $\overline{c}$ (black line) is the primary source of momentum within all three layers. As the intermediate destruction layer is approached from the near-bed region ($z^+ \approx 245$) the gradient of Reynolds stress decreases (solid red profile), while change in kinetic energy (blue profiles) and viscous diffusion (dashed red profile) increases their contribution. In the intermediate layer, the balance appears as a complex interplay between the different contributions. The most interesting balance is in the interface layer, where $\overline{c}$ is primarily balanced by the steady increase in streamwise kinetic energy, which occurs primarily through a slow diffusional thickening of the interface layer. The implication for long distance evolution of the current is that while the near-bed layer's height remains fixed, capped by the intermediate destruction layer, the momentum within the interface layer will slowly diffuse upward, as in a laminar Couette flow. The possibility of such a lower driving layer of constant thickness and self-similar velocity, driving an ever growing upper driven layer has been discussed by Luchi et al.[12]. As they point out, a large part of the suspended sediment is sequestered in the driving near-bed layer with only a smaller portion contained within the driven upper layer. Furthermore, this partition into driving and driven layer is not greatly altered at small settling velocity of sediments.

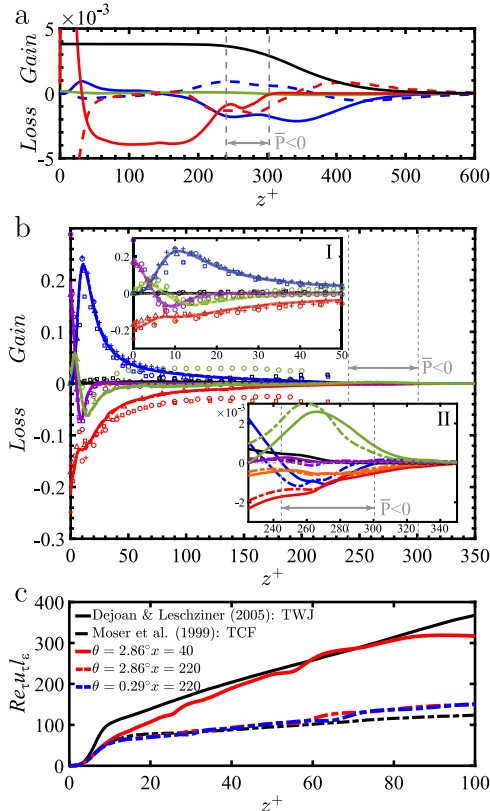

**Fig. 5 Scaled mean balances as a function of bed-normal location $z^+$.**
**a** Scaled mean streamwise momentum balance as a function of $z^+$ for the subcritical gravity current. (blue line), $-\frac{\partial \overline{u'^2}}{\partial x}$; (blue dashed line), $-\frac{\partial \overline{uw}}{\partial z}$; (green line), $-\frac{\partial \overline{p}}{\partial x}$; (red dashed), $\frac{1}{Re_\tau}\frac{\partial^2 \overline{u}}{\partial z^2}$; (red line), $-\frac{\partial}{\partial z}\overline{u'w'}$; (black line), $\overline{c}$. **b** Scaled mean TKE balance, as a function of $z^+$ for the subcritical gravity (solid profiles) and turbidity (dash-dotted profiles) currents. The profiles for the subcritical gravity and turbidity currents cannot be distinguished, except in the intermediate layer (inset II). (blue line), Production; (red line), Dissipation; (violet line), Viscous diffusion; (black line), Convection; (green line), Turbulent diffusion + Velocity-pressure gradient correlation; (orange line), Bed-normal Reynolds Flux. Also plotted are results from DNS of TWJ[30] (open circle), LES of TWJ[29] (open square), DNS of TBL[56] with cross (+), and DNS of TCF[51] (open triangle). Colors of each term in the data are the same as described above. Vertical dashed gray lines indicate locations of zero total TKE production. **c** Scaled turbulent length scale $Re_\tau u_\tau l_\varepsilon$ as a function of bed-normal location.

Finally we address the question of how different the turbulence is between the subcritical and the supercritical currents, by comparing them to a canonical turbulent boundary layer (TBL), turbulent channel flow (TCF) and turbulent wall-jet (TWJ). Figure 5b shows the following mean TKE balance as a function of bed-normal position $z^+$ within the subcritical gravity (solid profiles) and turbidity (dash-dotted profiles) currents:

$$\frac{1}{Re_\tau u_\tau^4}\left(\overline{P} - \overline{\varepsilon} + \frac{1}{Re_\tau}\frac{\partial^2 \overline{k}}{\partial z^2} - \frac{\partial \overline{T}}{\partial z} - \frac{1}{\tan\theta}\overline{w'c'} - \frac{\partial \overline{uk}}{\partial x}\right) = 0. \quad (2)$$

The terms on the LHS are (in order): production, dissipation, viscous diffusion, transport ($\overline{T} = \overline{w'(p' + \frac{1}{2}k)}$), bed-normal Reynolds flux and streamwise convection of TKE. Colors correspond to the different terms in the mean TKE balance. The profiles of the gravity and turbidity currents cannot be distinguished in the near-bed layer (inset I), although there is a clear difference in the intermediate layer (inset II). In the near-bed layer, general good agreement is observed with all the other numerical results (see

inset I). However, a closer inspection shows that above $z^+ \gtrsim 30$, dissipation and turbulent diffusion in the TWJ and TBL are larger than those of subcritical currents. On the other hand, we find excellent agreement for all the terms with DNS data of TCF[51]. This behavior is expected as high intensity turbulence in the interface layer influences the near-bed layer in a TWJ, while in the subcritical currents the intermediate destruction layer acts as a lid and the near-bed layer should be thought more as a TCF than a TBL. In the intermediate layer (see inset II), dissipation, production and Reynolds flux (orange profile) are balanced mainly by bed-normal turbulent and pressure diffusion $-\frac{\partial \overline{T}}{\partial z}$ (green profiles). Moreover, the bed-normal location for the intermediate layer decreases in the case of the turbidity current, as seen by the production profiles (blue) in inset II. In the interface layer, all terms in the TKE balance become negligible above $z^+ \approx 350$.

The arrested streamwise evolution of the near-bed layer in the subcritical currents can be further explored by computing the turbulent length scale $l_\varepsilon = \overline{k}^{3/2}/\overline{\varepsilon}$ in the near-bed layer. Figure 5c presents the normalized turbulent length scale $Re_\tau u_\tau l_\varepsilon$ for the subcritical and supercritical gravity currents as a function of $z^+$, together with the results of TWJ[29] and TCF[51]. Focusing on the supercritical gravity current (red profiles) we can see that close to the inlet ($x = 40$) the normalized turbulent length scale is comparable to that of TWJ, emphasizing the importance of highly turbulent interface layer. After the flow develops and stratification becomes dominant (after $x = 220$), the turbulent length scale decreases to values close to canonical TCF. This highlights the reduced interaction between the near-bed and interface layers as a result of stratification. Moreover, very similar values of turbulent length scale are observed between the subcritical and supercritical gravity currents. However, as evidenced by the location of streamwise velocity maximum $z_{max}$ (see Fig. 1e, f), the near-bed layer of a subcritical current behaves as a fixed free-slip lid, compared to a supercritical current whose near-bed layer continues to slowly grow in thickness.

In summary, the three-layer structure of the body of a subcritical current and the resulting very slow bed-normal growth offer an important mechanism for the long running body of turbidity currents in submarine channels. While there may be other possible scenarios of long runout, this work presents a detailed look at how the presence of an intermediate destruction layer near the streamwise velocity maximum decouples the turbulence of the near-bed layer from penetrating into the non-turbulent stably-stratified interface layer that forms a lutocline.

## Methods

**Numerical methodology.** The simulations from the present work are performed using a highly scalable, spectral element solver[52,53] using resolutions $336 \times 14 \times 44$ hexahedral elements with up to $16^3$ Gauss-Lobatto-Legendre (GLL) grid points. As a result, we use resolutions of up to 908 million grid points. The code solves the incompressible Navier-Stokes equations and the transport equation for the concentration[43]. The spectral element method exhibits small numerical dissipation and dispersion, which is important in obtaining statistically steady state solutions of turbulent flows, like in the present work[54]. For the gravity currents the domain size is $L_x \times L_y \times L_z = 96\pi \times 14 \times 20$, in the streamwise, spanwise and bed-normal directions, respectively. For the turbidity current the domain size is $L_x \times L_y \times L_z = 96\pi \times 8/3\pi \times 10$. Open boundary conditions are used at the top ($L_z$) and outflow ($L_x$) locations, which allow the unhindered evolution of the flow[55]. Moreover, periodic boundary conditions are used in the spanwise direction. At the inlet ($x = 0$), we use the statistically steady state solution of an auxiliary simulation of a turbidity current with a roof of height $2H$[46]. At the bottom boundary, a no-slip and zero net resuspension boundary conditions are used for the velocity and concentration fields, respectively. With this, the total amount of sediment in the current is conserved (bypass mode).

## Data availability

The simulation data that support the findings of this study are available in Open Science Framework with the identifier DOI 10.17605/OSF.IO/EZK2Y.

## Code availability

Source code is available at https://github.com/Nek5000/Nek5000. More information about the open source code can be found at https://nek5000.mcs.anl.gov/. Nek5000 is licensed under BSD.

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

## Acknowledgements

The authors would like to thank ExxonMobil for their support. The simulations were performed at UFHPC and we acknowledge their support. S.Z. and M.I.C. would also like to acknowledge CONICET, CNEA and UNCuyo.

## Author contributions

J.S., S.B., M.I.C., M.S., J.F., and D.H. designed research; J.S and S.Z. performed research; J.S. analyzed data; J.S. and S.B. wrote the paper.

## Competing interests

The authors declare no competing interests.
