## [Peer Review File · Nature Communications]

REVIEWER COMMENTS

Reviewer #1 (Remarks to the Author)

Overall: This contribution is proposing a new three-layer structure of the structure for subcritical turbidity currents, in which turbulence is destroyed within an intermediate layer near the velocity maximum, and with an upper layer which is only very weakly turbulent (as shown in panel D below). The proposal is based on numerical modelling, and comparisons to previous laboratory experiments, rather than say full field scale measurements.

Figure 1 (parts C and D) from manuscript showing new subcritical flow model in panel D.

Importantly, the modelling (and lab experiments) are simulating a very specific subset of flow types, which are entirely dilute, lack dense near-bed layers (which have different physics), and do not exchange sediment with the bed via erosion and deposition (as the flow is driven by excess sediment, these bed-exchange processes can totally dominate flow behaviour). The modelling also assumes either no sediment settling (effectively a 'saline flow'), or that sediment settling is very slow. This is not going to be the case in many oceanic flows. I understand why these key assumptions may indeed be needed for this type of modelling, but they are very fundamental – and need to be made crystal clear to all readers (including a general audience). As it stands, some key assumptions (and their wider implications) are not made clear enough to that general audience. However, this is an interesting proposal, in which the structure of subcritical flow is very different to that of supercritical flow, and it is underpinned by some very detailed and through modelling.

I think this is could become a valuable study, but it needs to address several key points for publication by Nature Geoscience. Some constructive suggestions are also now outlined.

(1) Clearer introduction that explains wider significant and novelty to a general (NG)

audience: There needs to be a much clearer introductory sections on what is really novel here, and why it matters; thus how this paper relates to (and goes beyond) past models for turbidity current structure. This section needs to be accessible to a wide scientific audience. As it stands, the manuscript rather comes across as being written for a very specific (i.e. flow modelling community) and needs to zoom out to place the new work in this wider context.

(1b) The work thus also needs to be placed in the wider context of past work.

Previous key work includes proposals that there is a 'turbulence minimum and barrier to momentum transfer' near the velocity maximum, linked to that reduction in velocity gradient and shear (e.g. by Kneller, Buckee and others), a later hypotheses (Lucchi, Parker et al.) that such a barrier can help explain a lack of mixing along the flow's upper boundary (such that flows

'mix themselves into huge thicknesses'), or more recent work by Dorrell et al. on 'sharpening' of that velocity maximum. It should also make clear how this proposed new model (panel D) is related to past saline laboratory experiments (by Sequiros et al.).

(1c) Clearer statement of novelty: My feeling is that the model (panel D) could actually be novel and very interesting, but this may just need to be articulated in a more accessible way to a wider scientific audience. The novelty may be around (i) the exact nature of that intermediate turbulence destruction zone (although a broadly comparable turbulence minimum zone/barrier has been proposed before), (ii) that the turbulence destruction zone only occurs in subcritical (and not supercritical) flows, and (iii) that an upper very weakly turbulent zone occurs in the subcritical flows (Fig. 2's panel D outer/interface layer - above).

Point (i) needs to say how this paper differs from past work, whilst points ii and iii rather surprised me (and I thus found them interesting). I wondered if a type of 'turbulence and momentum transfer barrier' also occurs in supercritical flows near their velocity maximum, albeit perhaps more weakly. I think past work suggests the barrier at the velocity maximum is likely to be a rather general property of entirely dilute turbidity currents. Point iii was the most surprising, and I wonder if this upper non-turbulent layer is partly a scaling issue; as flow speeds increase and there is more mixing with the ambient – this non-turbulent layer becomes vanishingly thin. If I have it right, this paper assumes that most turbulence is generated close to the bed, and then fails to be transferred above the velocity maximum. What happens when turbulence is generated by mixing along the upper surface of the flow, and thus does not need to travel through the velocity-maximum barrier? I am also always wary of comparisons to those saline flows, where there are no particles, and flow speeds are far slower than field-scale flows. Is this model only applicable to very slow flows, which will tend to have less vigorous mixing near their upper boundary.

(1d) Why this structure matters more widely: There needs to be a brief zoomed out explanation about why this new model (panel D) for turbidity current structure matters, beyond just the subgroup of people who numerically model such flows. Thus, why turbidity currents matter to a general NG audience. This is typically done at the start of NG papers. It might state the flows form the largest sediment accumulation, canyons and channels on Earth, or how they transfer carbon, nutrients and fresher water through oceans globally...

(2) This modelling simulates a very specific subset of flow types, and fundamental assumptions needs to be made crystal clear (including for a general NG audience).

(2a) Flow are entirely dilute sediment suspensions, and lacks dense near-bed layers: My understanding is this approach assumes that the turbidity current is entirely dilute. It does not capture the (very different) physics of dense ($\gg 10$ to 50%) dense layers that may occur (and fundamentally drive) the flow. This issue needs a clear and accessible discussion. It is particularly important because recent detailed flow monitoring in the ocean (e.g. Hughes Clarke, 2016; Paull et al., 2018) suggests that these dense layers can occur, whilst a large body of work on ancient outcrops had debated their existence, ever since Kuenen in the 1950s (see original references in talking et al., 2012). Direct monitoring of active flows may suggest that dense layers tend to occur near the flow front (Azpiroz et al., 2017; Simmons et al., 2020), in faster flows (front speeds > 1.5 m/s; Hughes Clarke, 2016; Paull et al., 2018, Heerema et al., 2020), whilst flows may eventually slow down and become entirely dilute as they runout into the deep sea. Thus, there needs to be a careful discussion of field evidence

for dense near bed layers, how those dense layers may be fast moving and drive the entire flow, how they may related to substrate erosion (which this modelling 'switches off'), if slower subcritical flows can also have a dense layer (or whether this is just not known yet)....etc. This is a key issue for this particular paper, and perhaps for the field of turbidity current flow modelling going forward.

Figure from recent work summarising direct evidence that a dense near-bed layer can be present, especially in faster turbidity currents. Paull et al. (2018) and Hughes Clarke (2016). These dense layers drive the flow, and are not simulated in this type of modelling.

(2b) There is no erosion or exchange of sediment with the bed: A second fundamental assumption made here is that the turbidity current does no exchange any sediment with the bed, such that sediment erosion is exactly balanced by deposition. This matters because sediment drives the flow, and past work shows (e.g. Traer et al. 2012) how erosion can totally dominate flow evolution, or indeed flow structure via formation of dense layers (as above). So this key assumption also needs more careful discussion, and it needs to be made crystal clear why this is a very particular situation being modelled here.

This is also where a general reader needs absolute flow speeds, not just dimensionless plots, to help us compare to field measurements, and assess whether a lack of erosion is likely.

(2c) Front to back (longitudinal) flow structure: The structure of a turbidity current will vary markedly from its front to body and then tail, as parameters such as velocity and density vary markedly (see Azpiroz et al., 2017; Simmons et al., 2020 or Paull et al., 2018 amongst others). It was not clear to me how the front to back (longitudinal) flow structure varies in these models. Is Figure 3 from the front or back of the flow, where is the front of the flow in Figure 4, and how long is the flow and its runout in km (see point 3).

(2d) Sediment does not settle or is slow settling (and hence very fine): Settling of sediment from a turbidity current can fundamentally affect its behaviour and internal structure. For example, if enough sediment settles out – the flow ceases to exist. In some of these model simulations, sediment settling is entirely switched off ($V=0$), whilst in other models it is not clear what settling velocities are in absolute terms, and how they relate to shear velocities,

and hence how sediment settling is being scaled here. This topic of scaling is critically important, not least because more laboratory experiments fail to realistically scale the balance between flow power and sediment settling (ws/u^*), whilst saline flows neglect it fully. I appreciate there are good reasons for choices made here, but this issue also needs to be discussed in greater detail, and made crystal clear to a wider (non-modelling) audience.

Thus, in a general sense, a set of critical assumptions made by this type of modelling need to be made much clearer, and in a balanced and accessible way (not hidden by ‘technical jargon’). It is fine to make these assumptions, and this does not matter for publication, but make the assumptions clear, and discuss why this is a very specific subset of flow types.

(3) Providing absolute values of velocities, concentrations (not just non-dimensionalised values): It would be very useful for a wider audience to understand what the absolute values are for flow velocity, thickness, grain sizes and settling velocity, sediment concentrations etc. Are the simulated flows weak (10 cm/s) or powerful (10 m/s), are they low (0.01% vol) or high (20% vol) sediment concentrations, are the flows very fine grained – say 10 microns – and then would the cohesive sediment gel or flocculate? The paper says these are ‘long’ runout flows – but is ‘long’ a 1 km or 1,000 km? All of these scale need to be related to absolute values for distance, speed, concentration and height etc. This makes the paper more accessible to a general reader, and provides a chance to compare more easily to field observations; and it makes some of the assumptions rather clearer....

(4) Is any comparisons to field data possible, as there is a danger of a closed loop: This paper compares models (which may significant assumptions), with small scale laboratory experiments that make similar assumptions (e.g. flow is dilute or even saline) and whose scaling (e.g. ws/u^*) is debateable (and see de Lueew et al., for further scaling discussion).

Thus, it is not at all clear that either the numerical modelling or the lab-scale experiments reproduce the key features of full-scale turbidity currents in global oceans or lakes. There is a need to compare and validate both numerical models and lab experiments against full-scale field observations, which are now becoming available.

One remaining issue is that sediment concentrations are very hard to measure directly, and this is needed to calculate flow density and hence Fr , and say whether flow is sub or supercritical. However, some recent flow monitoring work has constrained flow concentrations (Simmons et al., 2020), and preceding work had measured velocity profiles (Xu et al., 2014; Monterey Canyon and Azpiroz et al. (2017) and Simmons et al. (2020) in Congo Canyon. Is there any way that the model output can be compared to these full-scale field observations? It is unsure if some of these flows are subcritical or supercritical – but my feeling is that the velocity profile predicted in fig. 2d – is actually not seen. So the field data may not support the new model.

Simmons et al. 2020 provide estimates of bulk Richardson Number, which may be relevant.

(5) The format of Nature Communication allows additional figures, and provides the space needed for the authors to address these comments.

I hope the comments are useful. It is a very thorough and detailed piece of numerical modelling, which may just need to zoom out a bit, and make its assumptions (and their

implications) even clearer to a general (e.g. Nature Comms) audience. I think it can greatly increase its impact in this way, and it was a pleasure to read.

Best regards Pete

Specific and More Minor Comments:

Introduction: I think that lines 14-19 needs to be replaces with two short but reasonably comprehensive paragraphs about (1) what field observations (not models) suggests turbidity currents actually comprise, and (2) what types of turbidity currents can and cannot be modelled with this approach, and that a key assumption is that grain to grain interactions or excess pore pressures do not matter.

Line 14-19: *“Turbidity currents are jet or plume-like flows bounded by a sloping bed at the bottom and with a stagnant layer of lighter ambient fluid above. The higher density of the current compared to the ambient fluid, due to suspended sediments, propels the current forward. In the particular case of turbidity currents with washload sediment, where the inertial and settling effects of particles are negligible, the flow is analogous to conservative gravity currents that are driven by temperature or salinity differences.”*

For example, what is meant by ‘these currents’ on line 22. “These currents are characterized by a two layer structure”. Which sort of currents, and are they the ones being modelled?

Line 20-22. *“An important feature of turbidity currents to be discussed here is that the current and the ambient fluid are miscible and, as a result, the thickness of the current steadily increases as it flows downstream by entraining ambient fluid into it”.*

First, the few observations we have from real oceanic flows suggest their thickness is relatively constant (see Paull et al., 2018; Heerema et al., 2020), and Lucchi et al. show how sediment settling or ‘turbulence boundaries’ can limit these flow thickness increases.

What is Figure 1 aiming to do, as it just shows saline flow structure. The structure of sediment laden turbidity currents in the oceans may be very different (it probably is from recent monitoring). You need to make (very clear) that particle-laden turbidity currents, especially those with dense near bed layers, and which erode – could be very different to this. Also, how fast are the saline flows, as very slow overall speeds may be needed to get a significant non-turbulent layer at top of flow in subcritical flow

Line 44-46: *As a result, in subcritical currents, only the near-wall layer is vigorously turbulent, while the interface layer remains stably stratified [8].*

There is no field data to back this up (Simmons et al., 2020), and the upper layer can be both stably stratified and turbulent. I am not at all sure this upper zone of non-turbulent flows occurs in subcritical flows. Is the concentration really so uniform in the basal layer, which have grain sizes that are more abundant near the bed.

Line 64-65. *In contrast, it has been observed that turbidity currents can extend over thousands of kilometers within submarine channels [12,14]. If they were to entrain ambient fluid and increase in height as a TWJ, they could not have traveled such long distances.*

This is very simplistic, as the long runout distance could be achieved by due to bed erosion, which is neglected here, or by dense near bed layers, or by autosuspension.

Figure 2 – there are some remarkably high (> 50%) concentrations near the bed. What are the absolute units in this figure?

Section on key ingredients: There needs a preceding section to make this more accessible. Does the modelling from here on assume a very dilute flow, or....?

Based on this observation, we present three key ingredients that are necessary for a subcritical current to evolve along the streamwise direction without growing in thickness: (M1) the near-wall layer must behave like a turbulent channel flow (TCF), unlike the TBL-like behavior of a TWJ, (M2) the interface layer must remain stably-stratified and thus exhibiting only weak diffusional growth, and (M3) there must be a stable intermediate layer between the near-wall and interface layers that strongly suppresses any upward transport of near-wall turbulence into the upper interface layer.

What about settling of sediment (and detrainment and lowering of upper flow boundary) balances turbulent mixing? That will help to reduce flow thickness.

M1: Line 94-110: but these excellent laboratory experiments still have scaling issues....and no erosion of the bed etc. Importantly, the development of very dense near-bed layers is neglected.

M2: The condition for stability of a stratified shear layer is given in terms of gradient Richardson number as $Ri_g > 0.25$. This condition is not satisfied in the interface layer of a supercritical current and, as a result, the upper layer remains turbulent. This condition is satisfied in the upper layer of a subcritical current, thereby rendering it stably-stratified.

What is the independent evidence that Gradient Ri is > 0.25 in either layer – I did not follow. There is danger of circular reasoning, such that if it does not fit your model, it is not true?

II Simulation Details

The flow is sufficiently dilute and allows the use of Boussinesq approximation. We consider the limiting case of small, non-cohesive particles where inertial effects of the particles can be ignored, and we assume the sediment velocity to be equal to the local fluid velocity plus its still fluid settling velocity [30, 31].

This needs clarification. How dilute? What about particle collisions or pore pressure effects in very dense flows. Is sediment settling ever hindered, line 157-158 suggests not. What about bed erosion?

Line 159-164. Among the many simulations performed, results from three particular simulations will be highlighted: (i) subcritical gravity current with a bottom slope $\theta = 0.29^\circ$, inlet densimetric Froude number $Fr = 0.83$, and washload sediment of zero settling velocity $V_s = 0$, (ii) supercritical gravity current with $\theta = 0.86^\circ$, $Fr = 2.65$, and $V_s = 0$, and (iii) subcritical turbidity current with $\theta = 0.29^\circ$, $Fr = 0.83$, and $V_s = 10 \mu\text{m/s}$, which in the case of an intense current of height 10m will correspond to sediment particles of size ≈ 40 microns.

I needed some absolute values here, of flow thickness speed etc. That helps to compare against the field data.

I note that (i) and (ii) are simulations of a denser non-particulate fluid – as the settling rate of particles is zero. How realistic is this? It certainly needs to be made crystal clear. What is meant by ‘intense current’ – how fast exactly. What grain size does a settling velocity of 0.001 refer to, to first

Line 143 – how long is a ‘very long’ domain. Some absolute units needed.

No deposition or erosion allowed, which is a very unusual scenario. It has major implications on flow behaviour and structure can be dominated by exchange of sediment with the bed. :
Line 164-166. *In all the simulations the streamwise sediment ux is held constant. I.e., the current is in by-pass mode where local deposition is exactly balanced by sediment resuspension.*

We need to know more about the model runs in Figure 3. For example, is $V=0$ (hence the sediment cannot settle, and it is like a denser fluid?).

Can we please avoid too many TLAs... *the near-wall layer must behave like a turbulent channel ow (TCF), unlike the TBL-like behavior of a TWJ....*

Line 223-224.

The profiles for the subcritical gravity and turbidity currents are almost identical.

This lost me, can you explain.

Line 388: *However, it must be cautioned that the three-layer structure alone is not sufficient to ensure very long runout. In addition, the current must evolve in a near-zero net-depositional (i.e., bypass) mode [5], the current must be channelized within a submarine channel systems, and the bottom slope of the channel must remain shallow over long distances, in order for the current to remain in the subcritical regime.*

The lack of exchange of sediment with the bed is very unusual, and in real flows exchange of sediment with the bed will be dominant control on flow evolution. For example, persistent erosion will lead to long runout flows, which may accelerate.

Some references cited:

Azpiroz-Zabala, M., Cartigny, M.J.B., Talling, P.J., Parsons, D.R., Sumner, E.J., Clare, M.A., Simmons, S., Cooper, C., and Pope E.L., 2017. Newly recognised turbidity current structure can explain prolonged flushing of submarine canyons. *Science Advances*, 3, e1700200

Buckee, Kneller, and Peakall, Turbulence structure in steady, solute-driven gravity currents, *Particulate gravity currents*, 173 (2001).

Dorrell et al., 2019. Self-sharpening induces jet-like structure in seafloor gravity currents. *Nature Comms*.

Heerema, C.J., Talling, P.J., Cartigny, M.J., Paull, C.K., Bailey, L., Simmons, S.M., Parsons, D.R., Clare, M.A., Gwiazda, R., Lundsten, E., Anderson, K., Maier, K.L., Xu, J.P., Sumner, E.J., Rosenberger, K., Gales, J., McGann, M., Carter, L., Pope, E., and Monterey Coordinated Canyon Experiment (CCE) Team. 2020. What determines the downstream evolution of turbidity currents? *Earth and Planetary Science Letters*, v. 532, 116023. [10.1016/j.epsl.2019.116023](https://doi.org/10.1016/j.epsl.2019.116023).

Luchi, et al., 2018. Turbidity currents with equilibrium basal driving layers: A mechanism for long runout, *Geophysical Research Letters* 45, 1518.

Paull, C.K., Talling, P.J., Maier, K., Parsons, D., Xu, J., Caress, D., Gwiazda, R., Lundsten, E., Anderson, K., Barry, J., Chaffey, M., O'Reilly, T., Rosenberger, K., Simmons, S., McCann, M., McGann, M., Kieft, B., Gales, J., Sumner, E.J., Clare, M.A., and Cartigny, M.J.B., 2018. Powerful turbidity currents driven by dense basal layers. *Nature Communications*, NCOMMS-18-09895A. doi: 10.1038/s41467-018-06254-6

Simmons, S. M., Azpiroz-Zabala, M., Cartigny, M. J. B., Clare, M. A., Cooper, C., Parsons, D. R., Pope, E. L., Sumner, E. J., and Talling, P. J., 2020. Novel acoustic method provides first detailed measurements of sediment concentration structure within submarine turbidity currents. *Journal of Geophysical Research*. doi:10.1029/2019JC015904

Talling, P.J., Sumner, E.J., Masson, D.G., and Malgesini, G., 2012, Subaqueous sediment density flows: depositional processes and deposit types. *Sedimentology*, v. 59, p. 1937-2003.

Traer, M. M., G. E. Hilley, A. Fildani, and T. McHargue (2012), The sensitivity of turbidity currents to mass and momentum exchanges between these underflows and their surroundings, *J. Geophys. Res.*, 117, F01009, doi:10.1029/2011JF001990.

J.P. Xu, Octavio E. Sequeiros, Marlene A. Noble, Sediment concentrations, flow conditions, and downstream evolution of two turbidity currents, Monterey Canyon, USA, Deep-Sea Research I, <http://dx.doi.org/10.1016/j.dsr.2014.04.001>

Reviewer #2 (Remarks to the Author):

Summary of review (detailed comments below):

This is an outstanding paper, well written and illustrated. Using a highly resolved numerical model the authors show how turbulence is suppressed near the velocity maximum of a subcritical flow, enabling the stability and long-distance run-out of such flows. The illustrations, particular Figs 1, 2 and 4 are exceptionally well prepared and clear as required to illustrate such a complex process. Suggestions for possible improvements (particularly graphs which may need to be redrafted for publication quality) are indicated below.

The hypothesis is well established and the method to test it well defined and explained, including the provision of access to the code and illustrations of the model results (the latter could not be confirmed).

My main suggestion before I would recommend straight publication of the paper is that the authors could improve on the implications of the work for natural systems. The long-distance transport of sediment to the deep sea through submarine channels (with distances in excess of 1000 km in several cases) has been a fundamental issue for marine sciences, particularly the transport of relatively coarse sediment to the far outer edges of submarine fans. The implications of this study are huge, and the authors could easily expand on such implications by relating their results to field-scale channels/flows. Expanding on the broad implications of this work would certainly help reach a broader audience and enhance the impact of the paper.

Abstract: very clear and well written. Suggest you clarify the sentence "...preventing the near-wall turbulent structure from penetrating into the interface layer BY BACK SCATTERING VELOCITY FLUCTUATIONS INTO MEAN MOTION" (caps to highlight needed clarification - need to make this understandable to those less familiar with such technical jargon).

line 57: authors used again the qualifier 'vigorous' to describe the turbulence inside a subcritical current (line 45). Unclear what this is based on. If turbulence is "vigorous" in the subcritical case, what would be the adjective for supercritical currents? perhaps best to remove such unquantified adjective? or else provide support.

line 61: reuse of adjective 'vigorous'... perhaps could diversify? strong?

line 69: suggest change to MUST remain

line 70: ... it WOULD not be able

but please note you indicated sediments in washload condition above (line 17), so if the settling velocity is 'negligible' would you still really need 'vigorous' turbulence? are we dealing with the case of $V_s \sim 0$ or in the more general case? please be specific.

- Your overall problem statement is clear: need to understand the mechanism to sustain a turbidity current over a long distance. Your underlying assumption is that the flow must be subcritical otherwise entrainment would kill the flow. So you propose a 'hypothesis' that a multi-layered structure keeps the lutocline stable (which is what you plan to demonstrate). However, are there alternatives to the hypothesis? One issue that you may want to mention is that the long distance flow observed in nature are also associated with submarine channels, where the flow is laterally confined (a point you raise at the very end). Also, these channels tend to be sinuous - your model deals with a 'straight' flow. Here you could frame better what is the precise question and your overall approach to resolve it? Reference [13] proposes that the flow must be subcritical in order to travel >1000 km. But there are still very few direct observations of these flows. Perhaps a reference to the Zaire canyon (others?) where measurements have been made would strengthen your paper. If suitable measurements do not exist, perhaps you should mention what needs to be done to demonstrate in the field?

Section I (l. 75-139): This section in part anticipates the results of the model (as presented in

section III), perhaps this section could be shortened or cut significantly, bringing the discussion around TWJ/TCF with the discussion of the results (as in done in section III line 203-204). Could use a diagram to highlight the contrasts between TWJ, TCF... (perhaps expand on figure 1?) to assist conveying the concepts in a more succinct manner and help generalize your results.

Line 143: very long domain... how is this defined?

Also: what is the width of the simulation domain – specify in section II.

Line 164-166: It is unclear how you can force the current not to deposit/erode. Is this a B.C. in the model? How do you ensure these equilibrium conditions to occur given an 'artificial' inlet boundary condition? Obviously this is ok for $V_s=0$, but with sediment there will always be a tendency to erode/deposit, particularly near the inlet before the current reaches such equilibrium. Please explain.

Line 187: strictly speaking there is a thin zone of negative turb. production in supercritical current too.

Line 220-221: repeated from line 149, and inconsistent (vertical scale is half-height H or height h?) specify all scalings early and only once to avoid confusion.

Line 334: define $z+$, scaling?.

Line 348: would be good to clarify the implications this study brings beyond ref [15]

Line 356-357 are repeated from fig 5 caption. Don't need in text.

Starting in line 356... (looks like a 'conclusions' section but very short)...

the 'other' necessary conditions (lines 389-392) appear to come as an afterthought! if they are so important why not discuss them earlier. What are the limitations of the model and approach taken? Also, how does your model (3 layers) relate to the so-called auto-suspension concept (by H. Pantin)? Also, what are the conditions that can lead to the 'death' of a subcritical flow? lateral spreading at the end of a channel? or slope->zero?

Could you bring in some typical natural system and compare with the results of your model? for instance you mentioned diffusional growth of the lutocline interface - over a duration of hours/days (ref 13) and distance ~ 1000 km, would your model quantitatively explain such flows? such comparison could enrich your conclusions/implications.

Fig1: indicate dimensions in A,B (and C,D). Unclear if E,F refer to interpretation of experiment in A,B or to a numerical model (profiles are measured, model or conceptual?). Also indicate units of vert/hor. axes (meters, cm, normalized?) OK, explained in line 149... should indicate in caption.

Notice that Figs 1C-D are not cited in text. Still unclear whether 1E/F are based on model or measurement.

Meaning of hachured area in 1E/F. Does it mean that a supercritical flow also has a three-layer structure? In theory the structure is the same as a subcritical flow, no? please explain better.

Fig.2: Are properties (u' , w' , c_{bar} , etc) normalized? or else indicate units? also indicate axis units (m, cm?) Indicate in caption meaning of "CV, HP" in image 2F. Indicate this is result of numerical model or experiment. Spell out acronyms, tke - should be upper case?

Fig. 3: Describe source of experimental data. Describe reasoning behind displaying the $P<0$ and $z=1, 1.4, 2$ lines (why display absolute z in a scaled axis?). Could improve quality of graphs for publication.

Fig. 4: wow. It is a nice illustration of a complex process! indicate in B,C,D the meaning of the isosurfaces colors.

Fig. 5: dashed line in 5C plot cannot be distinguished in legend

Refs [3] and [30] are repeated.

Check title of ref [34] (should be $R_{\text{sub_Theta}}$)

check title for ref [35] ($Re_{\text{sub_tau}}$)

Letter to reviewers

Anatomy of subcritical submarine flows with a lutocline and an intermediate destruction layer

Reviewer 1

This contribution is proposing a new three-layer structure of the structure for subcritical turbidity currents, in which turbulence is destroyed within an intermediate layer near the velocity maximum, and with an upper layer which is only very weakly turbulent (as shown in panel D below). The proposal is based on numerical modelling, and comparisons to previous laboratory experiments, rather than say full field scale measurements.

Importantly, the modelling (and lab experiments) are simulating a very specific subset of flow types, which are entirely dilute, lack dense near-bed layers (which have different physics), and do not exchange sediment with the bed via erosion and deposition (as the flow is driven by excess sediment, these bed-exchange processes can totally dominate flow behaviour). The modelling also assumes either no sediment settling (effectively a ‘saline flow’), or that sediment settling is very slow. This is not going to be the case in many oceanic flows. I understand why these key assumptions may indeed be needed for this type of modelling, but they are very fundamental – and need to be made crystal clear to all readers (including a general audience). As it stands, some key assumptions (and their wider implications) are not made clear enough to that general audience. However, this is an interesting proposal, in which the structure of subcritical flow is very different to that of supercritical flow, and it is underpinned by some very detailed and thorough modelling. I think this is could become a valuable study, but it needs to address several key points for publication by Nature Geoscience. Some constructive suggestions are also now outlined.

We thank the reviewer for his very careful reading of the manuscript and providing many detailed comments on the manuscript. They were very useful in substantially improving the manuscript.

In the original manuscript, while we conveyed the details of the three layer structure, from the comments it is very clear that we did not convey the context of the present study nearly as well. The present study focuses on the long body of a long running turbidity current away from the head region, for example, similar to the one in Congo canyon discussed in Azpiroz-Zabala *et al.* In the revision we are now careful about the term “long runout”, as it can be used in two different ways. In the case of a short-running (surge-type) current, “long runout” indicates that the current travels hundreds of kilometers along the submarine canyon. Such a current is able to travel far by being erosional and thereby does not dilute itself to extinction like a plume or jet. In the case of a long-running current, “long runout” indicates not only the frontal-cell (or the head) of the current travels long distance, but also the current has a very long body that extends over hundreds of kilometers along the submarine canyon. The question being addressed in this paper is how does such a continuous long body remains from becoming too thick. For example, the body of a current of velocity 0.8 m/s running for 6 days will be about 440 km in length and an entrainment coefficient of 0.001 will imply that it thickness increases by 440 m. This is clearly not the case. The three layer structure of the subcritical current along its body, away from the front, provides a mechanism for much smaller entrainment and growth of the current. This offers a plausible explanation of the long body of a continuous long runout.

These points, were not well articulated in the earlier manuscript. This was amply clear from many of reviewer’s comments and questions. We profusely thank the reviewer in helping us sharpen our message for a wider audience. We acknowledge the reviewer now in the acknowledgment.

General Comments

1. Clearer introduction that explains wider significant and novelty to a general (NG) audience.

- (a) There needs to be a much clearer introductory sections on what is really novel here, and why it matters; thus how this paper relates to (and goes beyond) past models for turbidity current structure. This section needs to be accessible to a wide scientific audience. As it stands, the manuscript rather comes across as being written for a very specific (i.e. flow modelling community) and needs to zoom out to place the new work in this wider context.

We thank the reviewer for this and the subsequent related comments. We tried to be somewhat cautious in our original manuscript by sticking to the observational facts and refraining from statements that may be perceived as speculations or overstatements. Perhaps we were over cautious. Encouraged by the reviewer comments, we have thoroughly revised the introduction to better bring out the novelty of the work, relevance to past models and potential interest to broader scientific community. This we accomplish by following the subsequent comments of the reviewer.

- (b) The work thus also needs to be placed in the wider context of past work. Previous key work includes proposals that there is a ‘turbulence minimum and barrier to momentum transfer’ near the velocity maximum, linked to that reduction in velocity gradient and shear (e.g. by Kneller, Buckee and others), a later hypotheses (Lucchi, Parker et al.) that such a barrier can help explain a lack of mixing along the flow’s upper boundary (such that flows ‘mix themselves into huge thicknesses’), or more recent work by Dorrell et al. on ‘sharpening’ of that velocity maximum. It should also make clear how this proposed new model (panel D) is related to past saline laboratory experiments (by Sequiros et al.).

In the revised manuscript we connect the proposed three layer model with the past models of (i) reduction in velocity gradient and shear (Kneller, Buckee and others), (ii) lack of mixing at the upper boundary (Lucchi, et al.), (iii) sharpening of velocity maximum (Dorrell et al.) and (iv) the laboratory experiments of Sequiros et al.

- (c) Clearer statement of novelty: My feeling is that the model (panel D) could actually be novel and very interesting, but this may just need to be articulated in a more accessible way to a wider scientific audience. The novelty may be around (i) the exact nature of that intermediate turbulence destruction zone (although a broadly comparable turbulence minimum zone/barrier has been proposed before), (ii) that the turbulence destruction zone only occurs in subcritical (and not supercritical) flows, and (iii) that an upper very weakly turbulent zone occurs in the subcritical flows (Fig. 2’s panel D outer/interface layer - above). Point (i) needs to say how this paper differs from past work, whilst points ii and iii rather surprised me (and I thus found them interesting). I wondered if a type of ‘turbulence and momentum transfer barrier’ also occurs in supercritical flows near their velocity maximum, albeit perhaps more weakly. I think past work suggests the barrier at the velocity maximum is likely to be a rather general property of entirely dilute turbidity currents. Point iii was the most surprising, and I wonder if this upper non-turbulent layer is partly a scaling issue; as flow speeds increase and there is more mixing with the ambient – this non-turbulent layer becomes vanishingly thin. If I have it right, this paper assumes that most turbulence is generated close to the bed, and then fails to be transferred above the velocity maximum. What happens when turbulence is generated by mixing along the upper surface of the flow, and thus does not need to

travel through the velocity-maximum barrier? I am also always wary of comparisons to those saline flows, where there are no particles, and flow speeds are far slower than field-scale flows. Is this model only applicable to very slow flows, which will tend to have less vigorous mixing near their upper boundary.

We like the three points enumerated by the reviewer and we are delighted to note that the reviewer found the latter points to be new and interesting. First and foremost we have restructured the introduction to bring out these points early on as per reviewer's suggestion.

We completely agree with the reviewer that the existence of a layer near the velocity maximum where shear TKE production is negative is a general property of all shear flows with an asymmetric velocity maximum. It exists even in case of supercritical currents and turbulent wall jets. However, this is not the case when total turbulent production is considered, where terms other than shear production contributes to total TKE. In the case of supercritical currents, the region of negative total TKE production is negligibly small. In contrast, in the case of subcritical currents a substantial region exists where total TKE production is negative and this region is termed *intermediate destruction layer*. This distinction is important and it is clarified in section IV A.

When the interface layer becomes turbulent, it is not due to the seepage of near-bed turbulence across the velocity maximum. As pointed above, even in supercritical currents, there is a weak barrier to momentum exchange. The shear-layer turbulence is generated by shear instability in the upper layer, starting from destabilization of small perturbations that exists within the shear-layer. Thus, in the subcritical case, not only that turbulence is unable to penetrate from the near-bed layer to the shear-layer, but also the upper layer remains stable without local instability and onset of turbulence. These important points are now discussed in the revision.

The intensity of mean streamwise velocity and turbulence in the near-bed layer are about the same in both the supercritical and subcritical currents simulated. Thus the non-turbulent nature of the subcritical current's interface is not due to the flow being very slow. Nevertheless we admit that the Reynolds number of the flow being simulated are in the range of laboratory experiments and much lower than field conditions.

It must be clearly stated that the present work considers only the body of the turbidity current, away from the distinct frontal cell or head of the current. This important aspect of our work has not been properly communicated and this may be the source of several of reviewer's comments. In the revision we state this aspect of the present work clearly upfront and then reinforce at a few other places as necessary.

- (d) Why this structure matters more widely: There needs to be a brief zoomed out explanation about why this new model (panel D) for turbidity current structure matters, beyond just the subgroup of people who numerically model such flows. Thus, why turbidity currents matter to a general NG audience. This is typically done at the start of NG papers. It might state the flows form the largest sediment accumulation, canyons and channels on Earth, or how they transfer carbon, nutrients and fresher water through oceans globally...

As mentioned earlier, we had stayed too focused on the mechanistic details. It was a mistake. Following the reviewer's suggestion, and encouragement, we have expanded on the importance in the revision. We thank the reviewer for the encouragement.

2. This modelling simulates a very specific subset of flow types, and fundamental assumptions

needs to be made crystal clear (including for a general NG audience).

- (a) Flow are entirely dilute sediment suspensions, and lacks dense near-bed layers: My understanding is this approach assumes that the turbidity current is entirely dilute. It does not capture the (very different) physics of dense ($\gg 10$ to 50%) dense layers that may occur (and fundamentally drive) the flow. This issue needs a clear and accessible discussion. It is particularly important because recent detailed flow monitoring in the ocean (e.g. Hughes Clarke, 2016; Paull et al., 2018) suggests that these dense layers can occur, whilst a large body of work on ancient outcrops had debated their existence, ever since Kuenen in the 1950s (see original references in talking et al., 2012). Direct monitoring of active flows may suggest that dense layers tend to occur near the flow front (Azpiroz et al., 2017; Simmons et al., 2020), in faster flows (front speeds $> 1.5m/s$; Hughes Clarke, 2016; Paull et al., 2018, Heerema et al., 2020), whilst flows may eventually slow down and become entirely dilute as they runout into the deep sea. Thus, there needs to be a careful discussion of field evidence for dense near bed layers, how those dense layers may be fast moving and drive the entire flow, how they may related to substrate erosion (which this modelling ‘switches off’), if slower subcritical flows can also have a dense layer (or whether this is just not known yet)...etc. This is a key issue for this particular paper, and perhaps for the field of turbidity current flow modelling going forward.

This is a great point. In the revision, in the introduction, we now clearly mention the approximations of the numerical model. Foremost among them, our numerical simulations model the long body of the turbidity current, far from the energetic frontal cell or head of the current. Therefore, the head of the current that forms at the beginning of the simulation is allowed to travel downslope and exit the computational domain. The long body of the current that remains within the computational domain after this initial transient phase is investigated in detail, with particular attention to the nature of the interface layer and the entrainment of ambient fluid from above. We also mention that the flow in the body of the current is taken to be dilute and as a result the model does not account for the possible presence of a dense near-bed layer as observed in the field measurements. We also refer to these recent publications appropriately.

- (b) There is no erosion or exchange of sediment with the bed: A second fundamental assumption made here is that the turbidity current does no exchange any sediment with the bed, such that sediment erosion is exactly balanced by deposition. This matters because sediment drives the flow, and past work shows (e.g. Traer et al. 2012) how erosion can totally dominate flow evolution, or indeed flow structure via formation of dense layers (as above). So this key assumption also needs more careful discussion, and it needs to be made crystal clear why this is a very particular situation being modelled here. This is also where a general reader needs absolute flow speeds, not just dimensionless plots, to help us compare to field measurements, and assess whether a lack of erosion is likely.

This is again a valid point. This approximation has been clearly stated and discussed in the revision. The present simulations consider the long body of turbidity currents in the bypass mode. We want to emphasize that erosion and deposition are **not assumed to be zero** in the bypass mode. The only assumption is that erosion and deposition are in dynamic balance. Clearly this is an idealization and cannot be the case over the entire length of a real turbidity current. But this important simplification has

allowed us to study the structure of subcritical current without the added complexity of varying sediment concentration. We strongly believe that the mechanisms discussed in the present work with this simplification are relevant even in real currents in regions where the flow is in near bypass mode.

As suggested by the reviewer, in the revision, in the section on simulation details, we now present the dimensional values of several relevant quantities for a typical laboratory-scale application and then for a field scale turbidity current.

- (c) Front to back (longitudinal) flow structure: The structure of a turbidity current will vary markedly from its front to body and then tail, as parameters such as velocity and density vary markedly (see Azpiroz et al., 2017; Simmons et al., 2020 or Paull et al., 2018 amongst others). It was not clear to me how the front to back (longitudinal) flow structure varies in these models. Is Figure 3 from the front or back of the flow, where is the front of the flow in Figure 4, and how long is the flow and its runout in km (see point 3).

We apologize for the confusion. We now clearly explain the focus of the paper to be the structure of the body of a long-running subcritical current that exists after the passage of the energetic front of the current. We refer to the works suggested by the reviewer and clarify that the entire paper including figures 3 and 4 pertain to the long-running body of a subcritical current. Along the body of the current the computational domain employed in the simulations correspond to a length of about 7.3 km, under corresponding field conditions.

- (d) Sediment does not settle or is slow settling (and hence very fine): Settling of sediment from a turbidity current can fundamentally affect its behaviour and internal structure. For example, if enough sediment settles out – the flow ceases to exist. In some of these model simulations, sediment settling is entirely switched off ($V=0$), whilst in other models it is not clear what settling velocities are in absolute terms, and how they relate to shear velocities, and hence how sediment settling is being scaled here. This topic of scaling is critically important, not least because more laboratory experiments fail to realistically scale the balance between flow power and sediment settling (w_s/u^*), whilst saline flows neglect it fully. I appreciate there are good reasons for choices made here, but this issue also needs to be discussed in greater detail, and made crystal clear to a wider (non-modelling) audience.

In the revised text we present the settling velocity in dimensional terms for a typical laboratory-scale application that we provide as an example. We also discuss what the scenario would be in the case of an example real turbidity current.

- (e) Thus, in a general sense, a set of critical assumptions made by this type of modelling need to be made much clearer, and in a balanced and accessible way (not hidden by ‘technical jargon’). It is fine to make these assumptions, and this does not matter for publication, but make the assumptions clear, and discuss why this is a very specific subset of flow types.

We are in total agreement with this summary suggestion. In the revision, following reviewer’s suggestions listed above we clearly and in simple terms discuss all the key assumptions of the present numerical simulations.

- 3. Providing absolute values of velocities, concentrations (not just non-dimensionalised values): It would be very useful for a wider audience to understand what the absolute values are for flow velocity, thickness, grain sizes and settling velocity, sediment concentrations etc. Are the

simulated flows weak (10 cm/s) or powerful (10 m/s), are they low (0.01% vol) or high (20% vol) sediment concentrations, are the flows very fine grained – say 10 microns – and then would the cohesive sediment gel or flocculate? The paper says these are ‘long’ runout flows – but is ‘long’ a 1 km or 1,000 km? All of these scale need to be related to absolute values for distance, speed, concentration and height etc. This makes the paper more accessible to a general reader, and provides a chance to compare more easily to field observations; and it makes some of the assumptions rather clearer...

As suggested by the reviewer, in the revision, we now present the dimensional values of the relevant quantities for a typical laboratory-scale application. Due to the lower Reynolds number of the present simulations, the dimensional quantities are discussed first in the context of a laboratory-scale application. We then scale to a higher Reynolds number and discuss the corresponding values for a possible field observation.

4. Is any comparisons to field data possible, as there is a danger of a closed loop: This paper compares models (which may significant assumptions), with small scale laboratory experiments that make similar assumptions (e.g. flow is dilute or even saline) and whose scaling (e.g. w_s/u^*) is debateable (and see de Lueew et al., for further scaling discussion). Thus, it is not at all clear that either the numerical modelling or the lab-scale experiments reproduce the key features of full-scale turbidity currents in global oceans or lakes. There is a need to compare and validate both numerical models and lab experiments against fullscale field observations, which are now becoming available. One remaining issue is that sediment concentrations are very hard to measure directly, and this is needed to calculate flow density and hence Fr , and say whether flow is sub or supercritical. However, some recent flow monitoring work has constrained flow concentrations (Simmons et al., 2020), and preceding work had measured velocity profiles (Xu et al., 2014; Monterey Canyon and Azpiroz et al. (2017) and Simmons et al. (2020) in Congo Canyon. Is there any way that the model output can be compared to these full-scale field observations? It is unsure if some of these flows are subcritical or supercritical – but my feeling is that the velocity profile predicted in fig. 2d – is actually not seen. So the field data may not support the new model. Simmons et al. 2020 provide estimates of bulk Richardson Number, which may be relevant.

We would dearly like to compare the simulation results with field observations that are in the *long body* of subcritical and supercritical currents, away from the energetic front region. Only with such comparison, the simulation results can be considered fully validated. As of now we have done our best effort to compare against the field observations of Simmons *et al.* and Dorrell *et al.*. Again it must be cautioned that the present simulations employ a number of simplifying assumptions (listed above), which only approximately model a field condition. Most important among them being the modest Reynolds number of the simulation. Nevertheless, we hope that the present simulations, along with other recent efforts, will inspire more detailed field measurements for further comparison.

5. The format of Nature Communication allows additional figures, and provides the space needed for the authors to address these comments. I hope the comments are useful. It is a very thorough and detailed piece of numerical modelling, which may just need to zoom out a bit, and make its assumptions (and their implications) even clearer to a general (e.g. Nature Comms) audience. I think it can greatly increase its impact in this way, and it was a pleasure to read.

In the original manuscript we have tried to be brief and focus only on the main aspects of the three-layer structure. The reviewers comments have been detailed and most helpful. They

have made the revised manuscript far better.

Specific and More Minor Comments

1. Introduction: I think that lines 14-19 needs to be replaces with two short but reasonably comprehensive paragraphs about (1) what field observations (not models) suggests turbidity currents actually comprise, and (2) what types of turbidity currents can and cannot be modeled with this approach, and that a key assumption is that grain to grain interactions or excess pore pressures do not matter.

In the revised manuscript these sentences have been replaced along the lines suggested by the reviewer. In fact, these new paragraphs have offered us the opportunity to address the earlier comments as well.

2. Line 14-19: “Turbidity currents are jet or plume-like flows bounded by a sloping bed at the bottom and with a stagnant layer of lighter ambient fluid above. The higher density of the current compared to the ambient fluid, due to suspended sediments, propels the current forward. In the particular case of turbidity currents with washload sediment, where the inertial and settling effects of particles are negligible, the flow is analogous to conservative gravity currents that are driven by temperature or salinity differences.” For example, what is meant by ‘these currents’ on line 22. “These currents are characterized by a two layer structure”. Which sort of currents, and are they the ones being modelled?

Excellent point. In the original manuscript, we were referring to a supercritical current in line 22. This we see can be a source of confusion. With the rewriting of the revised manuscript this issue has been addressed.

3. Line 20-22. “An important feature of turbidity currents to be discussed here is that the current and the ambient fluid are miscible and, as a result, the thickness of the current steadily increases as it flows downstream by entraining ambient fluid into it”. First, the few observations we have from real oceanic flows suggest their thickness is relatively constant (see Paull et al., 2018; Heerema et al., 2020), and Lucchi et al. show how sediment settling or ‘turbulence boundaries’ can limit these flow thickness increases.

This is related to the earlier point. In the original manuscript, we were referring to a supercritical current in these lines. This we see can be a source of confusion. With the rewriting of the revised manuscript this issue has been addressed.

4. What is Figure 1 aiming to do, as it just shows saline flow structure. The structure of sediment laden turbidity currents in the oceans may be very different (it probably is from recent monitoring). You need to make (very clear) that particle-laden turbidity currents, especially those with dense near bed layers, and which erode – could be very different to this. Also, how fast are the saline flows, as very slow overall speeds may be needed to get a significant non-turbulent layer at top of flow in subcritical flow

Figure 1 describes the structure of the *body of the current* in the two different canonical regimes (subcritical and supercritical regimes) in the washload limit. We now point out that these models are not universal in their representation and that the profiles can be different under conditions such as near the head region or in the presence of a dense near-bed layer.

5. Line 44-46: *As a result, in subcritical currents, only the near-wall layer is vigorously turbulent, while the interface layer remains stably stratified [8].* There is no field data to back this up

(Simmons et al., 2020), and the upper layer can be both stably stratified and turbulent. I am not at all sure this upper zone of non-turbulent flows occurs in subcritical flows. Is the concentration really so uniform in the basal layer, which have grain sizes that are more abundant near the bed.

We again emphasize that these comments pertain to only the long body of a subcritical current, away from the energetic front of the current. In the present simulations, we observe the upper interface layer of a subcritical current to be non-turbulent and stably stratified, while the near-wall layer to be turbulent. This is indeed the main point of the three-layer structure. This behavior has been observed in the body of subcritical currents in laboratory experiments of Sequeros et al., which we refer to. Also, additional support can be drawn from event 8 of Simmons *et al.* (2020), which shows low mixing at the interface reflected in low ambient fluid entrainment. The non-turbulent nature of the subcritical interface layer has also been suggested in Kneller *et al.* 2016. If the interface layer had been turbulent, wouldn't it make the subcritical current to grow in height? Perhaps we did not fully comprehend the question.

6. Line 64-65. *In contrast, it has been observed that turbidity currents can extend over thousands of kilometers within submarine channels [12,14]. If they were to entrain ambient fluid and increase in height as a TWJ, they could not have traveled such long distances.* This is very simplistic, as the long runout distance could be achieved by due to bed erosion, which is neglected here, or by dense near bed layers, or by autosuspension.

The reviewer makes a valid point that was not properly explained in lines 64 and 65. In terms of sediment concentration, a fresh supply of sediment by erosion could clearly permit the head of a current to travel very long runout distances. As pointed out at the very beginning of our response, here we are interested in the question of how continuously fed currents with a long body exists without interfacial mixing and growth. For a current to extend over a length by several hundred kilometers, its entrainment must be substantially small. This point is now made clear in the revision

7. Figure 2 – there are some remarkably high ($> 50\%$) concentrations near the bed. What are the absolute units in this figure?

Concentration presented in Fig. 2 is non-dimensionalized with the inlet concentration c_v . As pointed out in the revised manuscript, a field equivalent of the present subcritical current would correspond to a sediment concentration of $\approx 0.5\%$, which is consistent with the assumption of dilute current made in our numerical model.

8. Section on key ingredients: There needs a preceding section to make this more accessible. Does the modelling from here on assume a very dilute flow, or...?

The restructuring of the introduction should now clearly state the assumptions including the assumption of dilute sediment concentration.

9. *"Based on this observation, we present three key ingredients that are necessary for a subcritical current to evolve along the streamwise direction without growing in thickness: (M1) the near-wall layer must behave like a turbulent channel flow (TCF), unlike the TBL-like behavior of a TWJ, (M2) the interface layer must remain stably-stratified and thus exhibiting only weak diffusional growth, and (M3) there must be a stable intermediate layer between the near-wall and interface layers that strongly suppresses any upward transport of near-wall turbulence into the upper interface layer"*

What about of settling of sediment (and detrainment and lowering of upper flow boundary) balances turbulent mixing? That will help to reduce flow thickness.

This is a very good point that has not been articulated in the manuscript. Yes, the settling velocity of the sediment will play a role in determining how the upper boundary of that sediment class will grow along the flow direction. I.e., if the settling velocity is larger than entrainment velocity there will be effective detrainment. But in a real turbidity current, where a wide range of sediments are present, for all sediment sizes to be non-growing, we require the worst case situation of wash load to be non growing as well. This leads to the three key ingredients.

10. M1: Line 94-110: but these excellent laboratory experiments still have scaling issues. . . .and no erosion of the bed etc. Importantly, the development of very dense near-bed layers is neglected.

We agree that this limitation applies to the laboratory experiments as well.

11. "M2: The condition for stability of a stratified shear layer is given in terms of gradient Richardson number as $Ri_g > 0.25$. This condition is not satisfied in the interface layer of a supercritical current and, as a result, the upper layer remains turbulent. This condition is satisfied in the upper layer of a subcritical current, thereby rendering it stably-stratified"

What is the independent evidence that Gradient Ri is > 0.25 in either layer – I did not follow. There is danger of circular reasoning, such that if it does not fit your model, it is not true?

This is an excellent point. In the revised manuscript we simply state our observations, without any causal reasoning. Furthermore, we have refined these statements to be more precise. (i) The condition for stability of a stratified shear layer is given in terms of gradient Richardson number as $Ri_g > 0.25$. (ii) Ri_g computed from the simulation results show that this condition is violated somewhat within the interface layer of a supercritical current and we observe the interface layer to be turbulent. (iii) This condition is satisfied everywhere within the interface layer of a subcritical current and we observe the interface layer to be non-turbulent.

Simulation Details

1. "The flow is sufficiently dilute and allows the use of Boussinesq approximation. We consider the limiting case of small, non-cohesive particles where inertial effects of the particles can be ignored, and we assume the sediment velocity to be equal to the local fluid velocity plus its still fluid settling velocity [30, 31]" This needs clarification. How dilute? What about particle collisions or pore pressure effects in very dense flows. Is sediment settling ever hindered, line 157-158 suggests not. What about bed erosion?

These details are provided in the revised manuscript. We consider only the body of a turbidity current where the sediment concentration has been taken to be dilute. This allows neglect of sediment-sediment interaction, hindered settling, pore-pressure effects, and so on. The actual concentrations of an experimental model as well as of a field observation that correspond to the present simulations are presented. As mentioned before, under the bypass mode assumption, rate of sediment erosion is equal to the rate of sediment deposition.

2. Line 159-164. "Among the many simulations performed, results from three particular simulations will be highlighted: (i) subcritical gravity current with a bottom slope $\theta = 0.29$, inlet densimetric Froude number $Fr = 0.83$, and washload sediment of zero settling velocity $V = 0$, (ii) supercritical gravity current with $\theta = 0.86$, $Fr = 2.65$, and $V = 0$, and (iii) subcritical

turbidity current with $\theta = 0.29$, $Fr = 0.83$, and $V = 10E - 3$, which in the case of an intense current of height 10m will correspond to sediment particles of size 40 microns” I needed some absolute values here, of flow thickness speed etc. That helps to compare against the field data.

As suggested by the reviewer, in the revision we now present the dimensional values of the relevant quantities for a typical laboratory-scale application. Due to the lower Reynolds number of the present simulations, the dimensional quantities are discussed first in the context of a laboratory-scale application. We then scale to a higher Reynolds number and discuss the corresponding values for a possible field observation.

3. I note that (i) and (ii) are simulations of a denser non-particulate fluid – as the settling rate of particles is zero. How realistic is this? It certainly needs to be made crystal clear. What is meant by ‘intense current’ – how fast exactly. What grain size does a settling velocity of 0.001 refer to, to first

As discussed above, the parameter values that correspond to these simulations are presented in the revised manuscript.

4. Line 143 – how long is a ‘very long’ domain. Some absolute units needed.

We have used computational domains whose length in the streamwise direction is up to 150 current heights. For a field current of height 48.4 m, given as an example in the revised manuscript, this corresponds to 7.3 Km. Thus we simulate a 7.3 Km segment of a much longer body of a subcritical current.

5. Line 164-166: *”In all the simulations the streamwise sediment flux is held constant. I.e., the current is in by-pass mode where local deposition is exactly balanced by sediment resuspension”* No deposition or erosion allowed, which is a very unusual scenario. It has major implications are flow behaviour and structure can be dominated by exchange of sediment with the bed.

We again emphasize bypass mode allows deposition and erosion to be present. It only assumes the rate of erosion to approximately balance the rate of deposition, so that the streamwise flux of sediment remains the same along the local length of the current. We acknowledge that this is a simplified scenario. However, we believe that this idealizations has played a key role in permitting us to focus on the key mechanisms and the three-layer structure of the subcritical current.

6. We need to know more about the model runs in Figure 3. For example, is $V = 0$ (hence the sediment cannot settle, and it is like a denser fluid?).

Fig. 3 shows velocity, concentration, Ri_g and Re_b profiles. The solid lines correspond to gravity current where $V = 0$ and the dashed lines correspond to turbidity current where settling velocity is non-zero.

7. Can we please avoid too many TLAs... the near-wall layer must behave like a turbulent channel flow (TCF), unlike the TBL-like behavior of a TWJ...

We have modified the sentence to minimize the use of TLAs.

8. Line 223-224. The profiles for the subcritical gravity and turbidity currents are almost identical. This lost me, can you explain.

For the small sediment size considered, the results of the turbidity current with non-zero sediment settling velocity are nearly identical to those of the simulation where the settling

velocity is taken to be zero. In Figure 3 we observe the results of the two simulations to be very similar. The main conclusion to be drawn is that although the results are presented for gravity current with zero sediment settling velocity, they apply equally for the body of a turbidity current with sediment particles of small settling velocity. This point is clarified in the manuscript.

9. Line 388: *"However, it must be cautioned that the three-layer structure alone is not sufficient to ensure very long runout. In addition, the current must evolve in a near-zero net depositional (i.e., bypass) mode [5], the current must be channelized within a submarine channel systems, and the bottom slope of the channel must remain shallow over long distances, in order for the current to remain in the subcritical regime"* The lack of exchange of sediment with the bed is very unusual, and in real flows exchange of sediment with the bed will be dominant control on flow evolution. For example, persistent erosion will lead to long runout flows, which may accelerate.

We again point out that bypass allows finite non-zero erosion and deposition. It only assumed the two to nearly balance. Thus over the length of the current considered the rate of deposition balances the rate of resuspension so that the net flux of sediment that enters the computational domain is the same as that exits the domain. We agree that there may be other mechanisms of long runout. This point is made in the revised manuscript.

References

- Azpiroz-Zabala, M., Cartigny, M.J.B., Talling, P.J., Parsons, D.R., Sumner, E.J., Clare, M.A., Simmons, S., Cooper, C., and Pope E.L., 2017. Newly recognised turbidity current structure can explain prolonged flushing of submarine canyons. *Science Advances*, 3, e1700200
- Buckee, Kneller, and Peakall, Turbulence structure in steady, solute-driven gravity currents, *Particulate gravity currents*, 173 (2001).
- Dorrell et al., 2019. Self-sharpening induces jet-like structure in seafloor gravity currents. *Nature Comms*.
- Heerema, C.J., Talling, P.J., Cartigny, M.J., Paull, C.K., Bailey, L., Simmons, S.M., Parsons, D.R., Clare, M.A., Gwiazda, R., Lundsten, E., Anderson, K., Maier, K.L., Xu, J.P., Sumner, E.J., Rosenberger, K., Gales, J., McGann, M., Carter, L., Pope, E., and Monterey Coordinated Canyon Experiment (CCE) Team. 2020. What determines the downstream evolution of turbidity currents? *Earth and Planetary Science Letters*, v. 532, 116023. 10.1016/j.epsl.2019.116023.
- Luchi, et al., 2018. Turbidity currents with equilibrium basal driving layers: A mechanism for long runout, *Geophysical Research Letters* 45, 1518.
- Paull, C.K., Talling, P.J., Maier, K., Parsons, D., Xu, J., Caress, D., Gwiazda, R., Lundsten, E., Anderson, K., Barry, J., Chaffey, M., O'Reilly, T., Rosenberger, K., Simmons, S., McCann, M., McGann, M., Kieft, B., Gales, J., Sumner, E.J., Clare, M.A., and Cartigny, M.J.B., 2018. Powerful turbidity currents driven by dense basal layers. *Nature Communications*, NCOMMS-18-09895A. doi: 10.1038/s41467-018-06254-6

Simmons, S. M., Azpiroz-Zabala, M., Cartigny, M. J. B., Clare, M. A., Cooper, C., Parsons, D. R., Pope, E. L., Sumner, E. J., and Talling, P. J., 2020. Novel acoustic method provides first detailed measurements of sediment concentration structure within submarine turbidity currents. *Journal of Geophysical Research*. doi:10.1029/2019JC015904

Talling, P.J., Sumner, E.J., Masson, D.G., and Malgesini, G., 2012, Subaqueous sediment density flows: depositional processes and deposit types. *Sedimentology*, v. 59, p. 1937-2003.

Traer, M. M., G. E. Hilley, A. Fildani, and T.McHargue (2012), The sensitivity of turbidity currents to mass and momentum exchanges between these underflows and their surroundings, *J. Geophys. Res.*, 117, F01009, doi:10.1029/2011JF001990.

J.P. Xu, Octavio E. Sequeiros, Marlene A. Noble, Sediment concentrations, flow conditions, and downstream evolution of two turbidity currents, Monterey Canyon, USA, *Deep-Sea Research I*,

Letter to reviewers

Anatomy of subcritical submarine flows with a lutocline and an intermediate destruction layer

Reviewer 2

This is an outstanding paper, well written and illustrated. Using a highly resolved numerical model the authors show how turbulence is suppressed near the velocity maximum of a subcritical flow, enabling the stability and long-distance run-out of such flows. The illustrations, particular Figs 1, 2 and 4 are exceptionally well prepared and clear as required to illustrate such a complex process. Suggestions for possible improvements (particularly graphs which may need to be redrafted for publication quality) are indicated below. The hypothesis is well established and the method to test it well defined and explained, including the provision of access to the code and illustrations of the model results (the latter could not be confirmed). My main suggestion before I would recommend straight publication of the paper is that the authors could improve on the implications of the work for natural systems. The long-distance transport of sediment to the deep sea through submarine channels (with distances in excess of 1000 km in several cases) has been a fundamental issue for marine sciences, particularly the transport of relatively coarse sediment to the far outer edges of submarine fans.

The implications of this study are huge, and the authors could easily expand on such implications by relating their results to field-scale channels/flows. Expanding on the broad implications of this work would certainly help reach a broader audience and enhance the impact of the paper.

We thank the reviewer for the very careful reading of the manuscript and providing many detailed comments on the manuscript. They were very useful in substantially improving the manuscript.

Comments

1. Abstract: very clear and well written. Suggest you clarify the sentence "...preventing the near-wall turbulent structure from penetrating into the interface layer BY BACK SCATTERING VELOCITY FLUCTUATIONS INTO MEAN MOTION" (caps to highlight needed clarification - need to make this understandable to those less familiar with such technical jargon).

This technical jargon was unnecessary and has been removed.

2. line 57: authors used again the qualifier 'vigorous' to describe the turbulence inside a subcritical current (line 45). Unclear what this is based on. If turbulence is "vigorous" in the subcritical case, what would be the adjective for supercritical currents? perhaps best to remove such unquantified adjective? or else provide support.

The qualifier was not necessary and has been removed, since it is clearly a source of confusion.

3. line 61: reuse of adjective 'vigorous'... perhaps could diversify? strong?

The qualifier was not necessary and has been removed.

4. line 69: suggest change to MUST remain

Done

5. line 70: ... it WOULD not be able but please note you indicated sediments in washload condition above (line 17), so if the settling velocity is 'negligible' would you still really need 'vigorous' turbulence? are we dealing with the case of $V_s \sim 0$ or in the more general case? please be specific.

The reviewer makes a good point. This sentence is intended to apply even under conditions when the sediment settling velocity is non-zero. This sentence has been reworded to convey the message clearly.

6. Your overall problem statement is clear: need to understand the mechanism to sustain a turbidity current over a long distance. Your underlying assumption is that the flow must be subcritical otherwise entrainment would kill the flow. So you propose a 'hypothesis' that a multi-layered structure keeps the lutocline stable (which is what you plan to demonstrate). However, are there alternatives to the hypothesis? One issue that you may want to mention is that the long distance flow observed in nature are also associated with submarine channels, where the flow is laterally confined (a point you raise at the very end). Also, these channels tend to be sinuous - your model deals with a 'straight' flow. Here you could frame better what is the precise question and your overall approach to resolve it? Reference [13] proposes that the flow must be subcritical in order to travel $> 1000km$. But there are still very few direct observations of these flows. Perhaps a reference to the Zaire canyon (others?) where measurements have been made would strengthen your paper. If suitable measurements do not exist, perhaps you should mention what needs to be done to demonstrate in the field?

The suggestion to compare and quote field observations is a good one and we have tried our best to do that in the revision. As of now we have done our best effort to compare against the field observations of Simmons *et al.* and Dorrell *et al.*. Such measurements are hard to come by, especially in the long running body of the current (and not near the front of the current) and in the subcritical regime. It would be great if some were to measure detailed vertical profiles of velocity and sediment concentration to evaluate if the three layer structure exists in real long running subcritical currents. This point is made in the revised manuscript.

7. Section I (l. 75-139): This section in part anticipates the results of the model (as presented in section III), perhaps this section could be shortened or cut significantly, bringing the discussion around TWJ/TCF with the discussion of the results (as in done in section III line 203-204).

We appreciate the comment. One of our motivations in choosing this journal was to reach out to non-fluid mechanics experts. The purpose of this section was to prepare the wider audience to what we plan to present in terms of technical details in the later section. We would like to keep it in this spirit. Also this section was appreciated by the other reviewer, and thus the view point may differ depending on the background.

8. Could use a diagram to highlight the contrasts between TWJ, TCF... (perhaps expand on figure 1?) to assist conveying the concepts in a more succinct manner and help generalize your results.

We seriously considered this suggestion. Our attempts at modifying figure 1, made it very complicated and did not bring out any additional value, since TWJ and TCF are well-studied canonical flows. So, we decided to leave figure 1 without these additions. If the reviewer has a specific idea on expanding figure 1, we will be delighted to implement.

9. Line 143: very long domain. . . how is this defined? Also: what is the width of the simulation domain – specify in section II.

We have performed simulations where the computational domain is 150 times as long as the current height at the inlet. The width of the channel is also defined in the revision.

10. Line 164-166: It is unclear how you can force the current not to deposit/erode. Is this a B.C. in the model? How do you ensure these equilibrium conditions to occur given an ‘artificial’ inlet boundary condition? Obviously this is ok for $V_s=0$, but with sediment there will always be a tendency to erode/deposit, particularly near the inlet before the current reaches such equilibrium. Please explain.

In the revision this is mentioned in the introduction: *”Finally, we assume the body of the turbidity current to be in bypass mode, where sediment erosion and deposition occur along the bed, but their rates nearly balance each other so that the streamwise flux of suspended sediment is a constant along the length of the current.”* From the numerical perspective, this boundary condition is achieved by setting the vertical concentration gradient representing the resuspension flux to balance the sedimentation flux. This leads to a Robin boundary condition, where the concentration gradient at the boundary is related to the concentration value. Thus the net exchange between the current and the bed is zero.

11. Line 187: strictly speaking there is a thin zone of negative turb. production in supercritical current too.

We agree that there is a small region of negative shear production in the supercritical case as well. However, total TKE production involves contributions other than shear production. We observe total TKE production to be negative only over a vanishingly small region in the supercritical case, whereas it remains negative over a substantial region in the subcritical case that forms the intermediate destruction layer. This point is clarified in the revision.

12. Line 220-221: repeated from line 149, and inconsistent (vertical scale is half-height H or height h ?) specify all scalings early and only once to avoid confusion.

Sorry for the confusion. The profiles in Figure 3 are normalized by sediment layer thickness h . While the simulations were performed with the half-height H at the inlet as the length scale. We have clarified this in the revision.

13. Line 334: define z^+ , scaling?.

We now define $z^+ = Re_\tau u_\tau z$.

14. Line 348: would be good to clarify the implications this study brings beyond ref [15]

We have expanded on the implication in the revised manuscript.

15. Line 356-357 are repeated from fig 5 caption. Don’t need in text.

We have removed this text.

16. Starting in line 356... (looks like a ‘conclusions’ section but very short)... the ‘other’ necessary conditions (lines 389-392) appear to come as an afterthought! if they are so important why not discuss them earlier.

We have removed the other necessary conditions from the conclusion section. In the revision, we have significantly modified the introduction. The assumptions and limitations of the present simulations are now spelled out. Thus, these points that are removed from the conclusion are indirectly embedded in the revised introduction.

17. What are the limitations of the model and approach taken?

This has now been discussed clearly in the revised introduction.

18. Also, how does your model (3 layers) relate to the so-called auto-suspension concept (by H. Pantin)?

One way to see this as the opposite of the auto-suspension mode, where the current accelerates in a run-away mode, whereas here the current reaches a fixed-point-like equilibrium. Of course, the big limitation we place on the simulation is that the total amount of sediment is conserved, whereas an auto-suspension current will be erosional. Recent field measurements of long running turbidity currents in the Congo canyon (Simmons et al. 2020) have found long running turbidity currents with a highly erodible front followed by a long, somewhat stable body (almost constant velocity). This novel picture would put our model in coexistence with the auto-suspension concept.

19. Also, what are the conditions that can lead to the 'death' of a subcritical flow? lateral spreading at the end of a channel? or slope \rightarrow zero?

Correct. These are two of the conditions that could lead to the death of a subcritical flow. Also, increased sediment settling velocity could result in turbulence suppression. Cantero et al. studied the necessary condition for turbulence suppression with a simplified model of turbidity current with a roof. They found that there exists a critical value $K_{crit} = (V/\tan\theta)_{crit}$ above which turbulence in the near-wall layer is suppressed. This could also lead to the death of subcritical flow.

20. Could you bring in some typical natural system and compare with the results of your model? for instance you mentioned diffusional growth of the lutocline interface - over a duration of hours/days (ref 13) and distance 1000 km, would your model quantitatively explain such flows? such comparison could enrich your conclusions/implications.

This point was raised by the other reviewer as well. In the revision, we now present the dimensional values of the relevant quantities for a typical laboratory-scale application. Due to the lower Reynolds number of the present simulations, the dimensional quantities are discussed first in the context of a laboratory-scale application. We then scale to a higher Reynolds number and discuss the corresponding values for a possible field observation.

21. Fig. 1: indicate dimensions in A,B (and C,D). Unclear if E,F refer to interpretation of experiment in A,B or to a numerical model (profiles are measured, model or conceptual?). Also indicate units of vert/hor. axes (meters, cm, normalized?) OK, explained in line 149... should indicate in caption.

We now clarify these points in the figure caption.

22. Notice that Figs 1C-D are not cited in text. Still unclear whether 1E/F are based on model or measurement.

Panels C and D are described in the results section. They are now explained in the figure caption as well.

23. Meaning of hachured area in 1E/F. Does it mean that a supercritical flow also has a three-layer structure? In theory the structure is the same as a subcritical flow, no? please explain better.

This is an excellent observation. The difference lies in the fact that there is a difference between the region of negative shear production of TKE and the negative total TKE production. I.e., there is a difference between hatched regions in figures 1E/F and regions within the white contours in figures 2A/B. This is clarified in the revised manuscript.

24. Fig.2: Are properties (u' , w' , \bar{c} , etc) normalized? or else indicate units? also indicate axis units (m, cm?) Indicate in caption meaning of "CV, HP" in image 2F. Indicate this is result of numerical model or experiment. Spell out acronyms, tke - should be upper case?

We mention in the text that the velocity is scaled by shear velocity at the inlet and concentration by sediment concentration at the inlet. We now present the meaning of CV and HP in the caption of Fig. 2. We also specify this is from numerical simulations. We spell out TKE in the revision.

25. Fig. 3: Describe source of experimental data. Describe reasoning behind displaying the $P < 0$ and $z=1, 1.4, 2$ lines (why display absolute z in a scaled axis?). Could improve quality of graphs for publication.

Experimental data is from Sequeiros et al. 2010, which is in the text and has been added to the caption. Displaying the region with negative total TKE production helps illustrate the different layers of the current in the context of familiar quantities, such as scaled velocity and concentration profiles of turbidity currents. Showing the values of $z=1, 1.4$ and 2 helps interpreting the locations where we show slices in Fig. 4. We have now improved the quality of plot.

26. Fig. 4: wow. It is a nice illustration of a complex process! indicate in B,C,D the meaning of the isosurfaces colors.

In the revision we indicate the meaning of the isosurface colors.

27. Fig. 5: dashed line in 5C plot cannot be distinguished in legend

The figure has been improved.

28. Refs [3] and [30] are repeated.

Corrected

29. Check title of ref [34] (should be R_{Θ})

Corrected

30. check title for ref [35] (Re_{τ})

Corrected

REVIEWER COMMENTS

Reviewer #1 (Remarks to the Author):

I thank the authors for making a thorough job of addressing the points raised in my initial review.

For example, clearer statements have been added to explain the wider significance of turbidity currents, including within the introduction, and this will help with the wider Nature Comms readership.

Some key assumptions have also been made even clearer to the reader, which is also really important. These assumptions include that the modelled part of the flow is entirely dilute, and lacks a dense near-bed layer; and that there is no net-erosion or deposition on the bed.

My feeling is still that these assumptions (especially the latter assumption) mean that a very specific type of turbidity current is being modelled here. This is because many turbidity currents on the seabed probably do exchange significant amounts of sediment with the seabed, and this exchange may indeed dominate how flows behave (as shown nicely by the modelling of Trear et al., 2012). Diverse field studies over the years show that more powerful turbidity currents can erode really significant amounts of sediment. For example, see Piper's study of the 1929 Grand Banks event, or recent work by Mountjoy et al. on the 2018 Kaikoura Canyon flow). Slower flows are often depositional, leading to thick deposits.

Thus I wondered if it would be good to explain to the reader not just which assumptions have been made, but also how often those key assumptions hold. Basically, not just alert the reader to which assumptions are made, but also discuss whether field flows follow those assumptions. But the main thing is simply to make sure that the reader fully understands the assumptions that are made, as now done.

The authors have also done a better job of explaining how this work related to previous studies by Lucchi, Kneller, Bucker and Dorrell, which proposed that there is a barrier to turbulence diffusion across the velocity maximum. This also helps to explain how this new paper extends that past work.

An even clearer and more direct statement of what exactly is novel here might still be useful. The key statements are a bit vague, such as 'elucidate on the structure' within the abstract. However, this looks like a useful advance in our understanding of the basic structure of subcritical turbidity currents, and thus worthy of publication here.

There are various minor (mainly editorial) suggestions on attached pdf. Some sentences may benefit from additional punctuation, such that they are more easily read.

In conclusion, I repeat my thanks to the authors for engaging fully with the initial comments, and thereby producing an even better contribution.

Reviewer #2 (Remarks to the Author):

Thanks for addressing my comments and improving the ms.

I have only a few minor text correction/suggestions:

21-22: "Turbidity currents form the largest sediment
22 accumulation, canyons and channels on Earth." CHANGE to:

"Turbidity currents are responsible for the formation of deeply eroded submarine canyons and channels that feed into giant deep-sea fans that represent the largest sedimentary accumulations on Earth." or something like it...

24: ecosystems (plural)

24-26: "They are responsible for massive

25 emplacement of sediment as turbidites, which with their large amounts of organic-matter

26 deposits now form the richest oil and gas reserves [3, 5]." suggest CHANGE TO:

"they are responsible for the emplacement of thick sand layers (turbidites) through geologic time which now form some of the richest oil and gas reservoirs."

Letter to reviewers

Anatomy of subcritical submarine flows with a lutocline and an intermediate destruction layer

Reviewer 1

Comments

I thank the authors for making a thorough job of addressing the points raised in my initial review. For example, clearer statements have been added to explain the wider significance of turbidity currents, including within the introduction, and this will help with the wider Nature Comms readership. Some key assumptions have also been made even clearer to the reader, which is also really important. These assumptions include that the modelled part of the flow is entirely dilute, and lacks a dense near-bed layer; and that there is no net-erosion or deposition on the bed.

The authors have also done a better job of explaining how this work related to previous studies by Lucchi, Kneller, Bucker and Dorrell, which proposed that there is a barrier to turbulence diffusion across the velocity maximum. This also helps to explain how this new paper extends that past work.

In conclusion, I repeat my thanks to the authors for engaging fully with the initial comments, and thereby producing an even better contribution.

We thank again the reviewer for his very careful reading of the manuscript and providing many detailed comments on the manuscript.

1. My feeling is still that these assumptions (especially the latter assumption) mean that a very specific type of turbidity current is being modelled here. This is because many turbidity currents on the seabed probably do exchange significant amounts of sediment with the seabed, and this exchange may indeed dominate how flows behave (as shown nicely by the modelling of Trear et al., 2012). Diverse field studies over the years show that more powerful turbidity currents can erode really significant amounts of sediment. For example, see Piper's study of the 1929 Grand Banks event, or recent work by Mountjoy et al. on the 2018 Kaikoura Canyon flow). Slower flows are often depositional, leading to thick deposits. Thus I wondered if it would be good to explain to the reader not just which assumptions have been made, but also how often those key assumptions hold. Basically, not just alert the reader to which assumptions are made, but also discuss whether field flows follow those assumptions. But the main thing is simply to make sure that the reader fully understands the assumptions that are made, as now done.

As stated in the manuscript, we have focused our attention on the long body of long running turbidity currents, far from the highly erosive head of the current. In the revision, right after stating the assumptions, we point out that the assumptions give the body of the current a specific character similar to that observed in the body of type 1 events presented in Simmons et al. 2020. We further point out that under conditions of strong erosion or deposition the structure and the dynamics of the body of the current can be different.

2. An even clearer and more direct statement of what exactly is novel here might still be useful. The key statements are a bit vague, such as 'elucidate on the structure' within the abstract. However, this looks like a useful advance in our understanding of the basic structure of subcritical turbidity currents, and thus worthy of publication here.

We again thank the reviewer for his observation. The last few sentences of the abstract now include a short summary of the key findings of our work (three-layer structure, where the intermediate layer acts as a barrier for momentum).

3. There are various minor (mainly editorial) suggestions on attached pdf. Some sentences may benefit from additional punctuation, such that they are more easily read.

Implemented these changes in revised manuscript.

4. Line 135: *"In a surge-type current, a long runout is possible provided the head of the current is sufficiently energetic to be net-erosional."*

Any current with an erosive head can do this - even week long flows. They do not need to be short lived surges.

We agree and we have reworded this sentence.

5. Line 269: *"The advantage of this assumption is that it greatly simplifies the boundary condition to be applied at the bottom of the computational domain and renders the streamwise sediment flux to be a constant along the streamwise segment of the body of the current being simulated."*

I think for balance you might also say that there is a disadvantage, as it makes this a very specific type of flow.

Again we agree. In the revision we now point out the specific nature of the present assumptions in response to earlier comment 1.

Letter to reviewers

Anatomy of subcritical submarine flows with a lutocline and an intermediate destruction layer

Reviewer 2

Thanks for addressing my comments and improving the ms. I have only a few minor text correction/suggestions.

We again thank the reviewer for his/her input and we are glad that the implementations of his/her suggestions in the previous revision were to his/her satisfaction.

Comments

1. 21-22: "Turbidity currents form the largest sediment accumulation, canyons and channels on Earth."
CHANGE to: "Turbidity currents are responsible for the formation of deeply eroded submarine canyons and channels that feed into giant deep-sea fans that represent the largest sedimentary accumulations on Earth." or something like it...
2. 24: ecosystems (plural)
3. 24-26: "They are responsible for massive emplacement of sediment as turbidites, which with their large amounts of organic-matter deposits now form the richest oil and gas reserves [3, 5]." suggest
CHANGE TO: "they are responsible for the emplacement of thick sand layers (turbidites) through geologic time which now form some of the richest oil and gas reservoirs."

Implemented suggestions in revised manuscript.

REVIEWERS' COMMENTS

Reviewer #1 (Remarks to the Author):

The authors are thanks for making a really thorough job of responding to all of the points raised.

Best regards

Pete Talling